# CMIP6 models overestimate sea ice melt, growth & conduction relative to ice mass balance buoy estimates

Alex E. West[1], Edward W. Blockley[1]

[1]Met Office Hadley Centre, FitzRoy Road, Exeter, EX1 3PB

*Correspondence to*: Alex E. West (alex.west@metoffice.gov.uk)

**Abstract.** With the ongoing decline in Arctic sea ice extent, the accurate simulation of Arctic sea ice in coupled models remains an important problem in climate modelling. In this study, the substantial CMIP6 model spread in Arctic sea ice extent and volume is investigated using a novel, process-based approach. An observational dataset derived from the Arctic Ice Mass Balance buoy (IMB) network is used to evaluate fluxes of melt, growth and conduction produced by a subset of

CMIP6 models, to better understand the model processes that underlie the large-scale sea ice states. Due to the sparse nature of the IMB observations, the evaluation is performed by comparing distributions of modelled and observed fluxes in the densely sampled regions of the North Pole and Beaufort Sea.

We find that all fluxes are routinely biased high in magnitude with respect to the IMB measurements by nearly all models, with too much melt in summer, and too much conduction and growth in winter, even as a function of ice thickness. We also

show that fluxes vary in ways which are physically consistent with the thermodynamic parameterisations used, and that these effects likely modulate the large-scale relationship between ice thickness and ice growth and melt in the CMIP6 models.

## 1 Introduction

Arctic sea ice has declined substantially over the satellite record (since 1979) both in terms of extent (Stroeve et al., 2012; Stroeve and Notz, 2018; Cai et al., 2021) and thickness (Kwok, 2018). Model projections from CMIP6 suggest that an ice-

free Arctic in summer is a likely occurrence within the next 10-30 years: however, models tend to underestimate the sensitivity of summer sea ice to global temperature increase (for example, Figure 1d of SIMIP Community, 2020). In addition, there is substantial variation in the present-day sea ice area and volume simulated by models from the CMIP6 ensemble. The causes are not yet well-understood, although Long et al. (2021) found sea ice extent simulation to compare better to reference datasets in models with higher spatial resolution and greater physical complexity, particularly from

December to June. Chen et al. (2023) found a similar, but very weak, association between model resolution and ice volume simulation accuracy relative to the PIOMAS forced ice-ocean model. The difficulty of finding clear associations between model complexity, and either ice volume or summer ice extent simulation accuracy, underlines the complexity of the processes driving sea ice evolution within the Arctic Ocean.

The mean state and future trend of Arctic sea ice are closely related, as annual mean ice thickness decreases more for thick
ice than for thin ice for the same increase in atmospheric forcing (Holland et al., 2006; Chen et al., 2023). This is due to the
thickness-growth feedback, by which thinner sea ice grows more quickly in winter (Massonnet et al., 2018). This negative
feedback is nonlinear, and operates more strongly as ice thickness approaches zero, opposing the direct effects of climate
warming. Reducing, but not fully negating, its effects, is the surface albedo feedback, a positive feedback operating over
larger scales, by which areas of lower average sea ice thickness melt more quickly during summer, due to albedo falling to
lower values sooner.

In fact, the sea ice volume is closely coupled to the seasonal ice growth and melt through these processes (West et al., 2022).
Seasonal ice growth and melt drive the sea ice volume evolution in an obvious way, but sea ice volume in turn modulates
how the ice growth and melt respond to thermodynamic forcing from above (atmospheric radiative fluxes, and near-surface
temperature and humidity), and from below (oceanic heat flux). A schematic view of this relationship is presented in Figure
1 of West et al. (2020). To understand the drivers of sea ice melt in response to long-term climate warming, it is necessary to
understand the evolution of this coupled system. Understanding the causes of variation in present-day Arctic sea ice area and
volume is therefore an important step towards reducing uncertainty in future projections.

Ideally, then, a full evaluation of Arctic climate in CMIP6 would include not just ice area and volume, but also ice growth
and melt; the forcing variables of surface radiation, temperature and humidity, as well as oceanic heat flux; and internal ice
thermodynamic quantities such as conduction. However, partly due to observational limitations, evaluation of Arctic climate
variables apart from sea ice in CMIP6 has been sparse. For example, Henke et al. (2023) evaluated surface temperature in a
subset of CMIP6 models relative to ERA5, and found a general cold bias. However, they noted that this could be caused by
observational inaccuracy, as reanalysed temperatures are known to be too warm over Arctic sea ice due to lack of surface
snow (Batrak & Müller, 2019).

Evaluation of the internal processes of the sea ice is in principle even more problematic, due to the extreme difficulties in
measuring these quantities. However, in West et al. (2020), a dataset of conduction and mass balance fluxes was constructed
from elevation and temperature data from the Arctic Ice Mass Balance buoy (IMB) network (maintained by the Cold
Regions Research and Engineering Lab, CRREL), and this dataset was used to evaluate the CMIP5 model HadGEM2-ES.
The evaluation produced results consistent with a previous surface radiation and sea ice study of the same model (West et al,
2019), confirming that ice growth, melt and conduction were all too strong in this model, causing the sea ice thickness
seasonal cycle to be too amplified. It also elucidated the sea ice simulation further by showing that the model's lack of
thermal inertia was likely causing winter ice growth to be too strong, a conclusion that would not have been possible without
the IMB evaluation.

The purpose of this study is to apply the same method to the CMIP6 ensemble; to perform a detailed evaluation of CMIP6
internal sea ice thermodynamics – the energy fluxes associated with melt, growth and conduction – relative to fluxes derived
from the IMB network using the methods described in West et al. (2020) and West (2021). This evaluation is restricted to a
subset of 17 CMIP6 models that provide all the relevant diagnostics, and is combined with a full evaluation of sea ice extent

& thickness, global & Arctic temperature, and surface radiative fluxes. Throughout this study the mass fluxes associated with ice melt and growth are treated as synonymous with the energy fluxes driving these, related by the specific latent heat of fusion of ice 3.35 x $10^5$Jkg$^{-1}$. Due to variations in ice salinity and density, the relationship between ice *volume* fluxes and energy fluxes is more complex for the IMBs (as discussed in West et al., 2020) and for one particular group of models, discussed in Section 4.2 of this study.

The study is set out in the following way. In Section 2 the models, IMB data, and other reference data, are introduced. In Section 3 the climate states of the models are described by evaluating sea ice extent and other climate variables. In Section 4 fluxes of melt, growth & conduction are evaluated with respect to the IMBs. In Section 5 this evaluation is extended further to show how these fluxes vary with ice thickness & snow depth, and here we attempt to account for the sampling biases inherent in the IMB data. In Section 6 conclusions are presented.

## 2 Models and data

### 2.1 CMIP6 models

The CMIP6 data request gave scope for a much larger set of sea ice diagnostics than for previous projects (Notz et al., 2016). In particular, diagnostics of the sea ice heat and mass budgets were requested, including full components of the sea ice mass balance and of the energy balance at the top and basal surfaces of the snow-ice column. However, not all models provided all or any of these diagnostics. In some cases but by no means all, this was because sea ice components were sufficiently simple that they would not have been meaningful. In order to be included in this study, a model would provide the following diagnostics: sea ice area, thickness or volume, top and basal melting flux, and top and basal conduction flux. 17 models were identified that provided all of these diagnostics (Table 1), representing contributions from 9 separate modelling centres. Hereafter this subset of CMIP6 models is referred to as the 'CMIP6subset'.

The sea ice components of models in the CMIP6subset share many common features. All use a sub-grid ice thickness distribution (ITD), in which ice in each grid cell is divided into distinct thickness categories. For each category, temperatures and energy fluxes are computed separately. The ITD is important because it allows for rapid ice growth at low thicknesses to be properly captured (Holland et al., 2005; Massonnet et al., 2019). All models in the CMIP6 subset allow sea ice to display thermal inertia, with multiple vertical layers of ice and at least one separate layer of snow simulated for each model. Thermal inertia is likely also important in achieving realistic basal conduction and hence winter ice growth (West et al., 2020). All models parameterise ocean-to-ice heat flux in similar ways, using schemes derived from McPhee et al. (1992).

Despite these common features, the models also differ considerably, and can be grouped, in terms of thermodynamic characteristics. Here we describe these characteristic groups, which are referred to frequently during the paper. The first two groups contain models from multiple institutions; the remaining groups correspond to single institutions.

1.  *GSI8.1 group.* This group comprises the five CSIRO-ARCCSS, MOHC and NIMS-KMA models, which all use the GSI8.1 sea ice configuration (Ridley et al., 2018). This configuration uses version 5.1.2 of the Los Alamos sea ice

model CICE, and features multilayer thermodynamics with a fixed salinity profile (Bitz and Lipscomb, 1999). It is notable for its lack of solar radiation penetrating into ice; all incident shortwave radiation is either reflected or absorbed at the surface. Melt ponds are modelled using the topographic scheme of Flocco et al. (2012).

2.  *Mushy-layer group.* This group comprises the four NCAR and NCC models, which use a different configuration of CICE5.1.2. In this configuration, penetrating solar radiation is modelled, and in addition salinity is fully prognostic,
with a 'mushy' liquid-ice layer at the base of the ice (Turner and Hunke, 2015); as with the GSI8.1 models, melt ponds are explicitly modelled, using a level-ice rather than topographic scheme (Hunke et al., 2013).

3.  *CMCC group.* The two CMCC models use a different configuration of CICE again, this time with CICE version 4; penetrating solar radiation is modelled, but salinity is prescribed. Melt ponds are simulated using the level-ice scheme of Holland et al. (2012), a simpler version of that used by the mushy-layer model group.

4.  *IPSL group.* The two IPSL models use the LIM3 sea ice model; these models are distinguished by a salinity scheme of intermediate complexity, in which a linear profile is derived from a prognostic bulk salinity (Boucher et al., 2020). They use only two vertical ice layers, but penetrating solar radiation is permitted. They do not simulate melt ponds explicitly, but model their effect on shortwave radiation through a parameterisation of albedo based on surface temperature.

5.  *MRI 'group' (one model only).* MRI-ESM2-0 uses a custom-built sea ice model with a single ice and snow layer, a fixed salinity profile and penetrating solar radiation permitted, based on Mellor & Kantha (1989). Melt ponds are parameterised using a similar framework to that of the IPSL group.

6.  *CNRM-CERFACS group.* The three CNRM-CERFACS models use GELATO6, a sea ice model with salinity, thermodynamics and melt pond treatment similar to LIM3, but with nine vertical ice layers instead of two (Voldoire
et al., 2019).

It is important to note that a model's sea ice simulation is not entirely or even mostly controlled by the characteristics of its sea ice component. The forcing received from the ocean and especially the atmosphere component control the sea ice simulation to first order (e.g. Olonscheck et al., 2019). The atmosphere and ocean components of the CMIP6 subset are also shown in Table 1. More than half of the models feature the same ocean component (NEMO3.6). Models with identical sea
ice components tend to use closely related atmosphere components: for example the GSI8.1 group all use UM version 10.6 (GA7.1 configuration, Walters et al., 2017) while the mushy-layer group use atmospheric components derived from CAM6 (Danabasoglu et al., 2020).

## 2.2 The IMB data

An ice mass balance buoy (IMB) is a collection of instruments frozen into a sea ice floe (Richter-Menge et al., 2006). An IMB typically consists of acoustic sounders to measure ice surface and base elevation, a thermistor string to measure ice and

snow temperatures at 10cm vertical resolution, and a datalogger to record and transmit data. Some also include air temperature and sea level pressure sensors, but these variables will not be examined in this study. Since 1993, 110 IMBs have been deployed in the Arctic Ocean by the Cold Regions Research and Engineering Laboratory (CRREL), mainly in the North Pole and Beaufort Sea subregions (Figure 1). Individual buoys have been analysed to provide useful process studies of e.g. ocean heat flux (Lei et al., 2014). We note that since 2015 many IMBs have also been deployed by other institutions, notably in the course of the MOSAiC experiment (e.g. Koo et al., 2020). However, due to the complexity of data processing, we do not attempt to enlarge the dataset used in this study relative to that used in West et al. (2020).

In West et al. (2020), IMB data was systematically analysed to produce a dataset of monthly mean ice melt, growth and conduction fluxes for the North Pole and Beaufort Sea regions that was used to evaluate ice thermodynamics in a CMIP5 model, HadGEM2-ES, identifying a number of model biases as described in the Introduction. The processing of the raw IMB data is described fully in West et al. (2020), but is briefly summarised again here. IMB raw timeseries of temperature or surface / interface / base elevation, measured at irregular times, are interpolated or binomially averaged in time to create regular timeseries which are then used to create timeseries of sea ice thickness and snow depth (West, 2020c). Fluxes of melt and growth are calculated from the elevation and temperature timeseries. Fluxes of conduction and heat storage are calculated from temperature gradients, using a reference layer 40-70cm above the ice base for the basal conduction as gradients are very weak at the ice base (West, 2020d). The IMB dataset contained around 500 data points for each analysed flux, and displayed seasonal and spatial variability consistent with observational and theoretical understanding of the Arctic Ocean climate. Due to the sparseness of the data, interannual variability could not be detected. Uncertainty in the derived fluxes due to ice salinity, conductivity and density was quantified, in addition to uncertainty due to choice of the reference layer used to calculate basal conductive fluxes. While uncertainty due to measurement error was not quantified, the values identified by Lei et al. (2014) of 0.01m and 0.1K for elevation and temperature measurement respectively imply uncertainties over an order of magnitude lower than those identified for the factors listed above.

## 2.3 Other observational datasets used in this study

Other Arctic climate variables besides ice energy fluxes are evaluated in Section 3 below, and the datasets used are described here. For ice area, we use HadISST.2 (Titchner and Rayner, 2014). For surface temperature, we use GISTemp v4 (GISTemp team, 2024; Lenssen et al., 2019), HadCRUT v5 (Morice et al., 2020), NOAAGlobalTemp v 6.0.0 (Huang et al., 2024) and Berkeley Earth (Rohde and Hausfather, 2020), with these four datasets used to characterise the plausible range of observational uncertainty.

For ice thickness, we use the PIOMAS forced ice-ocean model reanalysis (Schweiger et al., 2011) and Cryosat-2 radar altimetry (Kurtz and Harbeck, 2017). To characterise observational uncertainty in ice thickness, we use a bootstrapping method trained on the brief period of overlap of CryoSat-2 with our assessment period, 2011-2014, to generate for each region and month a plausible distribution of the discrepancy between PIOMAS and CryoSat-2. These were used to derive ranges of ice thickness for each month of the year, as well as annual mean ice thickness and seasonal cycle amplitude. It is

hoped that the use of CryoSat-2 ameliorates any bias arising from the use of PIOMAS, which itself contains a sea ice model like many evaluated here, as a reference dataset.

In addition, surface radiative fluxes are evaluated in Section 3, with respect to the ERA5 analysis (Hersbach et al., 2023) and CERES-EBAF (Loeb et al., 2009). We do not attempt to explicitly characterise observational uncertainty in these variables.

## 3 Sea ice and Arctic climate state evaluation


In this section, the sea ice state (area and thickness) simulated by the CMIP6 subset models is evaluated. Throughout we restrict evaluation to the Arctic Ocean region (Figure 1). The evaluation period chosen is the last 30 years of the CMIP6 historical simulations, 1985-2014.

Because ice expansion in the Arctic Ocean region is largely limited by the Eurasian and North American continents, most

inter-model variation in ice area occurs in the summer (Figure 2a). Six models achieve minimum ice area in August (the GSI8.1 models, plus NorESM2-MM); all others, like HadISST.2, achieve minimum area in September. In September, the highest area occurs in MOHC UKESM1.0-LL and NIMS-KMA UKESM1.0-LL ($5.9 \times 10^6$ km$^2$). The lowest ice areas occur in the two CMCC models ($0.0 \times 10^6$ km$^2$), but these are outliers: the next lowest ice area, of $2.6 \times 10^6$ km2, occurs in NCAR CESM2. Most models simulate year-round ice area that is either like, or much lower than, that suggested by HadISST.2.

There is considerable spread in annual mean ice thickness (Figure 2b), with the thickest ice in MOHC UKESM1.0-LL (annual mean 3.3m) and the thinnest ice in the CMCC models (annual mean 0.5m). The PIOMAS and CryoSat-2 observational references sit roughly in the middle of the model range with PIOMAS displaying an annual mean thickness of 1.7m. All models achieve maximum ice thickness in either April (CMCC & CNRM-CERFACS models) or May; PIOMAS also achieves maximum ice thickness in May but CryoSat-2 is much earlier, in March, although given problems in retrieving

sea ice thickness measurements during the summer months (e.g. as described in Tilling et al., 2018) it is not clear whether or not this indicates a general model inaccuracy. Minimum ice thickness is achieved in September in both PIOMAS and CryoSat-2, and for all models except CMCC-ESM2 and MRI-ESM2-0, which achieve minimum in October.

We define annual ice growth and melt as the difference in mean Arctic Ocean sea ice thickness between the seasonal maximum in April-May and the seasonal minimum in September-October. This quantity is highest in IPSL-CM6A-LR at

1.46m and lowest in NorESM2-MM at 1.00m (Figure 2c), whilst the value for PIOMAS is 1.15m. While annual ice growth and melt is in theory strongly influenced by the annual mean ice thickness via the surface albedo and thickness-growth feedbacks, these quantities are only weakly negatively correlated across the ensemble, with a correlation coefficient of -0.27. The CMCC and CNRM-CERFACS models are instrumental in this lack of correlation, displaying both low annual mean sea ice thickness and low annual ice growth/melt; without these models, the correlation is -0.79. For the CMCC models, the lack

of growth/melt is likely related to the complete loss of sea ice in many parts of the Arctic during July/August; a possible reason for the CNRM-CERFACS models is discussed in Section 4 below. Among the other models, the GSI8.1 and the IPSL

models tend to lie on a higher curve than the mushy-layer and MRI models (ice growth/melt is higher for a given annual mean ice thickness); the two smaller model groups display correlation of -0.93 and -0.90 respectively. Compared to the models, ice growth/melt from PIOMAS is relatively low: it estimates ice growth/melt similar to that of the CMCC/CNRM-CERFACS models, but annual mean ice thickness is most similar to that of NCAR CESM2-WACCM.

There is strong correlation (0.81) between annual mean ice thickness and September ice area amongst the CMIP6 subset (Figure 2c). This correlation remains strong when the outlier CMCC models are removed (0.79).

We compare the annual mean ice thickness to the anomaly in global 2m air temperature relative to the 1850-1899 average (Figure 3a), as well as to 2m air temperature anomaly averaged over the Arctic Ocean region (Figure 3b). There is strong correlation between 2m air temperature over the Arctic Ocean and ice thickness (-0.85). Correlation between global temperature and ice thickness is also quite high at -0.79. We evaluate these using for global temperature the four datasets described in Section 2 to represent observational uncertainty, and for Arctic Ocean ice thickness PIOMAS and Cryosat-2. Models tend to simulate greater ice thickness for a given Arctic Ocean and global temperature anomaly than is suggested by observations, which may be related to model tendency to underestimate sea ice response to a given rise in global temperature (Notz et al., 2020).

We finally evaluate surface radiative fluxes relative to the ERA5 reanalysis and to CERES-EBAF satellite dataset (Figure 4). Surface radiative fluxes are important for the Arctic sea ice seasonal cycle as they are the principal driver of the surface flux variation over sea ice (and hence ice melt and growth). The CMCC models are notably distinct from all other models, displaying higher summer net SW fluxes (due to upwelling SW differences, Figure 4a) and higher absolute LW fluxes in both directions during autumn (Figure 4b); both are close corollaries of these models' very low summer sea ice cover. The mushy-layer models tend to display lower downwelling SW fluxes in summer and higher downwelling LW fluxes in spring and early summer, suggesting more extensive cloud cover in these models. The GSI8.1 models, and to a lesser extent the IPSL models, model lower absolute LW fluxes in both directions during the cold season; for the MOHC models this bias was noted in West (2021) and is likely related to low liquid water fraction in clouds.

Annual mean net SW and net LW are anticorrelated across the CMIP6 subset (Figure 4d) such that total net radiative flux varies little between models, with all but the CMCC models averaging between -0.5 and 6.4 W m$^{-2}$; the average for ERA5 and CERES-EBAF is 0.5 and 1.5 W m$^{-2}$ respectively. The CMCC models show far more net SW, and hence net radiation, than is indicated by ERA5 and CERES-EBAF; the mushy-layer models show somewhat more net LW and less net SW. All other models lie quite close to ERA5 and CERES-EBAF.

It is likely that the total net surface flux, unlike the net radiative flux, is net upwards, with turbulent fluxes over open water areas in winter providing much of the additional negative component. For example, Table 1 of Winkelbauer et al., 2024 shows Arctic Ocean average net surface flux to be upwards in the vast majority of CMIP6 models.

## 4 Mass balance and thermodynamics evaluation

In this section, modelled fluxes of ice growth and melt, and of vertical conduction at the ice surface and base, are evaluated with respect to the IMBs. All evaluated fluxes are available as direct model diagnostics; the only processing required is to divide melt and growth fluxes by ice area fraction. This is because these fluxes are produced as grid box means, averages over both sea ice- and open water-covered areas; for greater comparability with the IMB values, they must be converted to ice-only means by dividing out ice area fraction. The conduction fluxes are produced in their raw form as means over ice, and do not require this processing step.

Throughout this section, modelled fluxes are compared and evaluated for the comparatively well-sampled North Pole and Beaufort Sea subregions shown in Figure 1 using the equivalent fluxes derived from IMBs. Fluxes from model points within these regions are collected into distributions, similar to the distributions derived from the IMBs but with many more data points. All model statistics are computed from these distributions using weighting by both grid cell area and by ice concentration. Similarity of distributions is assessed using a Welch t-test, and differences are judged significant at the 5% level.

### 4.1 Melt and growth fluxes

*Top melting*

The IMB measurements show top melting fluxes to be near-zero outside the summer months, and to achieve their maximum in July, at 23 and 41 W m$^{-2}$ in the North Pole and Beaufort Sea subregions respectively (Figure 5a,b). Most models reproduce this shape, except for the CNRM-CERFACS models which have a greatly delayed seasonal cycle, displaying their seasonal maximum in September. Of the remaining models, the CMCC models display the highest top melting fluxes in both regions, reaching around 100 W m$^{-2}$ in July; CSIRO-ARCCSS ACCESS-CM2 is next highest, again in both regions. At the lower end of the distribution, NorESM2-MM displays the lowest maximum at 15.8 and 42.2 W m$^{-2}$ in the North Pole and Beaufort Sea regions respectively; it is the only model whose maximum is below the IMB average, in the North Pole region only, although the difference is not significant.

Model distributions significantly different to those of the IMBs are indicated in Figure 5 by boxplots with bold lines and black triangles for means; those not significantly different have boxplots with fainter lines and green triangles for means. All models save for NCC NorESM2-MM are either biased high relative to, or not significantly different to, the observations. Apart from the CNRM-CERFACS models, which are phase-shifted, and the CMCC models, which are the highest, there is a partition between the GSI8.1 models and the mushy-layer models, with the former tending to display much higher top melting fluxes than the latter. Among the remaining models, the IPSL models tend to lie closer to the GSI8.1 groupand MRI-ESM2-0 closer to the mushy-layer group. With a handful of exceptions, in most regions and months the GSI8.1 and IPSL models are biased significantly high with respect to the IMBs, while the mushy-layer models and MRI-ESM2-0 are not.

*Basal melting*

Basal melting is small in the IMB measurements outside the months June-September (Figure 5c,d), although unlike for top melting a small number of nonzero winter basal melting fluxes occur in the North Pole region which includes warmer waters at the Atlantic sea ice edge. They display maximum basal melt in August, contrasting to the July maximum for top melt. From June-September, average IMB basal melting fluxes are 7.8, 12.8, 18.6 and 6.5 W m$^{-2}$ in the North Pole region, and 10.9, 27.3, 32.3 and 20.6 W m$^{-2}$ in the Beaufort Sea region. Most models reproduce the shape of this seasonal cycle; the CNRM-CERFACS models are phase-shifted earlier in the season in the North Pole region particularly, but the phase-shift is not as severe as for the top melt. The CMCC models display much greater basal melt than the other models, with CMCC-CM2-SR5 reaching 374 W m-2 in the Beaufort Sea region in August. In most cases the CMCC models do not report data for September as there is essentially no ice left in this month.

Among the other models, IPSL-CM6A-LR-INCA displays the highest maximum basal melting in the North Pole region (51.6 W m$^{-2}$) while NCAR CESM2 is highest in the Beaufort Sea region (70.8 W m$^{-2}$). NIMS-KMA UKESM1-0-LL and MOHC UKESM1.0 display the lowest maxima for the respective regions (12.5 and 17.2 W m$^{-2}$). In contrast to the top melt, the GSI8.1 models tend to display among the lowest basal melting fluxes, with ACCESS-CM2 a notable exception. This is consistent with the finding of Keen et al. (2021) that the portion of melt attributable to top melt is much higher in models that do not allow penetrating shortwave radiation.

In the North Pole region, basal melting in the two UKESM1-0 models is biased moderately low relative to the IMBs, while the remaining MOHC models and the NCC models are not significantly different to the IMBs. All other models are biased high relative to the IMBs in this region. In the Beaufort Sea region the two UKESM1-0 representatives remain biased low, but many of the models which are biased high tend to overlap more with the IMBs than is the case in the North Pole region. For example, the quartiles of ACCESS-CM2 are nearly identical to those of the IMB distribution, but a smaller number of very high fluxes cause the mean to be much higher and the distribution to be significantly different.

*Basal growth*

Basal growth fluxes are near-zero during the summer in the IMBs but increase in magnitude during the mid-late autumn and early winter to reach their greatest magnitude of -25.5 and -14.1 W m$^{-2}$ in February, or the North Pole and Beaufort Sea regions respectively (Figure 5e,f). The North Pole timeseries is somewhat noisy in winter, with much lower values in February than the other cold season months. The CMCC and CNRM-CERFACS models display greatly enhanced basal growth fluxes relative to the IMBs, with their highest magnitude exceeding -40 W m$^{-2}$, and in some cases approaching -100 W m$^{-2}$; the CNRM-CERFACS models in addition display a severe phase lead (5 months) with minima occurring in September. The remaining models produce minima in a similar range to the IMBs in the North Pole region, but this is largely due to the lower IMB values in February; in December and January most models are biased low, with only UKESM1-0-LL, MRI-ESM2-0 and the two NCC models similar to the IMB distribution. In the Beaufort Sea region all models are biased low

relative to the IMBs (MRI-ESM2-0 barely). Aside from the CMCC and CNRM-CERFACS models, the CSIRO-ARCCSS, IPSL and NCAR models tend to be biased most severely, with minima in the region of -30 W m$^{-2}$.

There is a consistent phase lead of 2-3 months among the non-CNRM-CERFACS models, with all of these attaining their minimum in November or December rather than February. Associated with this is a severe model bias in the autumn: the CMCC and CNRM-CERFACS models aside, the mean and standard deviation of the modelled autumn basal growth flux is -15.8 ± 3.8 W m$^{-2}$ in the North Pole region; this compares to -3.0 W m$^{-2}$ in the IMBs. This is likely to be caused at least partially by a sampling bias in collecting the IMB measurements: in the autumn, the strongest ice growth tends to occur in

thin, newly forming ice. For two reasons, this ice is undersampled by the IMBs. Firstly, IMBs are preferentially placed in thicker ice floes, as these are easier to access and enhance the expected survival period of the IMB. Secondly, while models report fluxes from an Eulerian perspective that automatically includes contributions from all sea ice within a grid cell, IMBs report from a Lagrangian perspective that inherently biases the sampled ice thickness distribution towards ice floes undergoing slower thickness changes. Ice floes that grow rapidly do not tend to do so for long, as the thickening ice weakens

the vertical temperature gradient and hence heat loss. An IMB that happens to measure fast ice growth remains trapped in the same ice floe, and will not subsequently measure growth in new areas. This sampling bias was investigated in West et al. (2020) and West (2021), and found to contribute partially to, but not completely explain, the model biases identified in these studies. In Section 5 below it is assessed again in the context of the biases identified for the CMIP6 models.

## 4.2 Conduction and heat storage

*Top conductive flux*

  Top conductive flux represents the downwards flux of energy from the surface of the snow-ice column into the snow/ice interior; the sign convention used throughout is that downwards fluxes are positive. The IMBs measure top conductive flux to be strongly upwards in winter, representing heat lost to the atmosphere, and weakly downwards in summer (Figure 6a,b). From October-March, the average flux is between -18 and -21 W m$^{-2}$ in the North Pole region (with minimum in December)

and between -12 and -19 W m$^{-2}$ in the Beaufort Sea region (with minimum in November). The summer maximum, achieved in July, is 4.4 and 2.9 W m$^{-2}$ in the two regions respectively.

  All models simulate somewhat stronger top conduction in winter than measured by the IMBs, showing more negative values, representing greater heat loss to the atmosphere. This is more marked in the Beaufort Sea than North Pole region. For example, January average top conductive flux in the North Pole region ranges from -21 W m$^{-2}$ (MOHC UKESM1-0 LL) to -

315 39 W m$^{-2}$ (CMCC-CM2-0); in the Beaufort Sea region the average ranges from -26 to -50 W m$^{-2}$ (the same models at the extremes). Models with smaller annual mean ice thickness tend to display greater conduction in winter, and vice versa. The relationship between thickness and conduction is evaluated explicitly in Section 5 below.

  Another notable feature is the five distinct clusters of models formed by the differing seasonal cycle shapes and amplitudes. The CMCC models display a very amplified seasonal cycle with maxima in excess of -100 W m$^{-2}$ for both models and

320 regions, consistent with their very low annual mean ice thickness. The CNRM-CERFACS models are not particularly

amplified, but again display a phase-shift, with minima occurring around April. The remaining models display seasonal cycles more similar to the IMBs, but with smaller differences between them. The GSI8.1 models form a distinct cluster with the highest summer conduction fluxes (biased high relative to the IMBs), possibly due to the lack of solar penetration: full SW absorption at surface causes a warmer surface and more conduction into the interior. These models also display amongst the weakest winter conduction (most similar to the IMBs). The mushy-layer models of NCAR and NCC are similar to the GSI8.1 models in winter but display negative conduction in summer (heat is conducted upwards from the interior to the top surface), indicating that these models warm the ice interior more rapidly than to the GSI8.1 models, possibly partly because they allow solar radiation to penetrate the ice. The remaining models, MRI-ESM2-0 and the two IPSL models, are most similar to the IMBs in summer, with a mixture of weakly positive and weakly negative fluxes, but display somewhat stronger conduction in winter than the IMBs and the GSI8.1 & mushy-layer models.

*Basal conductive flux*

Basal conductive flux represents the downwards flux from the ice interior into the ice base, driven by heat loss from the ice surface but modulated by the ice heat capacity, it is the principal driver of winter ice growth. As for top conduction, the basal conductive flux is shown as positive downwards throughout this section; during the winter, when conduction tends to occur upwards from the warm ocean to the cold atmosphere, conductive fluxes are usually negative. During the summer, basal conduction fluxes are usually small; inter-model variation in basal melting tends to be driven by oceanic heat flux rather than by the basal conductive flux; summer variations in oceanic heat flux are in turn mainly driven by variations in direct solar heating of the ocean (Maykut & McPhee, 1995; Steele et al., 2010; Keen and Blockley, 2018).

The IMBs' basal conductive fluxes are similar to top conduction but tend to be smaller in magnitude and with the seasonal cycle shifted slightly later in the year (Figure 6c,d). In both regions the maximum upwards flux occurs in January, at -14 to -15 W $m^{-2}$. Fluxes rise sharply towards zero between April-June, becoming very weakly positive in July and August. Mean fluxes fall sharply to negative values again in November (North Pole) and October (Beaufort Sea).

For the basal conductive flux, as for the top conductive flux, the CMIP6 subset form five qualitatively distinct model clusters. The CMCC models display the characteristic exceptionally high amplitude, with values in excess of -100 W $m^{-2}$ occurring in October for all region and model combinations; they also display high positive fluxes in summer, reaching 20 to 40 W $m^{-2}$ in July. The CNRM-CERFACS models also have elevated amplitude in winter, reaching -40 to -60 W $m^{-2}$ in the North Pole region but again approaching or exceeding -100 W $m^{-2}$ in the Beaufort Sea region; they also display a small phase lead, with the highest negative mean values occurring in September. However fluxes do not turn positive in summer.

The 'mushy-layer' models (from NCC and NCAR) display much smaller basal conduction fluxes in winter than all other models and the IMBs, in most cases not exceeding -5 W $m^{-2}$. This is due to other terms in the basal heat balance not reported in CMIP6 diagnostics (Hunke, E., personal communication). Notably, because in the mushy-layer scheme ice is formed at much lower density, a given energy flux is able to produce a much greater increase in ice thickness, with associated increased shallowing of the temperature gradient, and reduced conduction. While this may in part represent a real-world

effect, the counterpart to this 'missing' energy is the energy released during internal freezing of brine pockets within sea ice, which is not obviously reported in the CMIP6 diagnostics. Plante et al. (2024) discuss additional problems with the standard mushy-layer formulation, in particular noting that the congelation growth flux does not include contributions from the full energy loss at the base of the ice, resulting in the imbalance being transferred to frazil formation instead. These issues, and the IMB evaluation, suggest that these models' basal conductive fluxes may be biased low in magnitude during the freezing season.

The mushy-layer models also display higher positive fluxes in summer than all but the CMCC models (denoting strong conduction to the ice base from the ice interior), in many cases exceeding 10 W m$^{-2}$. Among the remaining models, the GSI8.1 models display behaviour distinct from that of the other non-GSI8.1 models (the MRI and IPSL models). Firstly, the GSI8.1 models have smaller (i.e. less negative) winter fluxes, closer to the IMBs. Secondly, the GSI8.1 models, like the CNRM-CERFACS models, continue to simulate weakly negative fluxes in summer, whereas the MRI and IPSL models simulate positive values similar to the IMBs.

*Heat storage flux*

The heat storage flux is calculated as the top conductive flux minus the basal conductive flux, and represents the rate of change of heat content of the snow-ice column; negative represents ice cooling, positive ice warming. The IMBs show maximum ice warming rates in May and June (5-7 W m$^{-2}$). In the North Pole region, maximum ice cooling occurs in October at -19 W m$^{-2}$, but in the Beaufort Sea region, the heat storage term has few points and high standard deviation at this time of year. This may be related to the dataset being dominated by thinner ice for which the calculation of basal conductive flux 40-70cm above the ice base is less valid.

As with the conductive fluxes the IMB models form five clusters with their own distinctive behaviour. The GSI8.1 and MRI/IPSL clusters are the most qualitatively similar to the IMBs, with cooling from September – March and warming from April – August; all but MRI-ESM2-0 show more cooling than in the IMBs during winter, while the GSI8.1 models show much stronger sensible heating in summer than MRI/IPSL and the IMBs. The mushy-layer models show a similar seasonal cycle shape, but translated downwards: stronger cooling in winter, and no warming in summer. This is likely related to the additional terms in the basal energy balance mentioned in the basal conduction evaluation above.

The CMCC models show similar cooling to the IMBs in winter but are very different in summer, showing stronger cooling than in winter (basal conduction higher than top conduction). The CNRM-CERFACS models show an even more curious seasonal cycle, with strongest ice cooling in April and May but with exceptionally strong ice warming in September and October driven by the strongly negative basal conduction flux.

## 4.3 Aggregate metrics

We compute the seasonal total top and basal melt for each model by multiplying, for each model, month and region, the top plus basal melting by the ice area in that region, and then summing over months of the year. We compute a similar metric for

the IMBs by multiplying the average top melt flux, plus the average basal melt flux, estimated for each month and region by the total ice area for that month and region, obtained from HadISST.2. For the rest of this subsection, the top plus basal melt is shortened to 'total melt', despite the fact that in both models and reality there is another process causing melt not evaluated here (lateral melting). The total melt flux thus obtained is scattered against annual mean ice thickness (squares in Figure 7a,c) and against seasonal cycle amplitude (squares in Figure 7b,d), with observed values derived from PIOMAS and CryoSat-2 as described in Section 2.3. The top melt flux is scattered separately in both figures (stars). In a similar way we compute the seasonal total congelation growth flux, shortened to 'total growth flux' despite not including the frazil growth term, and scatter these against the same variables (triangles).

When plotted against annual mean ice thickness (Figure 7a,b) a rough inverse relationship is seen: models with higher thickness tend to see less growth/melt, and vice versa. This is expected due to the ice thickness-growth, and albedo feedbacks, discussed in more detail below. The IMBs lie on the lower boundary of the scatter (in terms of magnitude) for both total melt and growth, although not for the top melting alone.

The relationship between ice thickness and top melt alone is weaker, partly because of the effect of the GSI8.1 models which have among the highest ice thicknesses but also have proportionately greater top melting. Correlation between ice thickness and top melt is stronger within model groups; for example, in the North Pole region, within the GSI8.1 and mushy-layer clusters correlation is -0.69 and -0.99 respectively, compared to -0.51 across the ensemble as a whole. The picture in the Beaufort Sea region is similar.

One effect of weighting monthly fluxes by ice area is that the 'high flux' models of CMCC and CNRM-CERFACS become less extreme relative to the other models and to the IMBs. This is because the highest fluxes in these models occur in months when ice area is relatively low, hence do not represent an exceptionally large amount of ice volume loss or gain (as a high flux is spread over a relatively small area). Nevertheless these models still report the highest total melt and growth fluxes.

When plotted against the amplitude of the ice thickness seasonal cycle a positive correlation is seen. The correlation is not perfect for several reasons: missing processes not evaluated here cause additional ice growth or melt (such as lateral melting and frazil ice growth); spatial correlation between ice concentration and melt & growth fluxes causes discrepancies when averaging; and any ice growth and melt terms which occur in the same month are effectively invisible to the ice thickness seasonal cycle. This last effect is extreme for the CNRM-CERFACS models, as discussed further in Section 4.4 below. In the Beaufort Sea region the observed relationship between ice growth and melt from the IMBs, and ice thickness amplitude from PIOMAS/Cryosat-2, is notably inconsistent with the models. This is likely another symptom of the ice thickness sampling bias.

## 4.4 Model group discussion: the influence of thermodynamic parameterisation and mean climate state

We briefly discuss the simulations of each model group in turn, linking the vertical energy fluxes to the Arctic climate variables evaluated in Section 3. The CMCC models are distinguished by very high melt, growth and conduction fluxes, but in each case the seasonal cycles are of a similar shape and phase to the IMBs. The high fluxes arise naturally in response to

the unusually low annual mean ice thickness of the CMCC models. Thinner sea ice supports a steeper temperature gradient between the (relatively) warm ocean and cold atmosphere in winter, and hence also grows more quickly in winter. On a large scale, thinner ice also melts more quickly in summer due to the ice albedo feedback. Hence the large biases in the CMCC fluxes reflect the sea ice state bias, to which the sea ice thermodynamics are responding in a physically realistic way.

The CNRM-CERFACS models also have amplified maxima and minima in melt, growth and conduction variables, but many also display large phase offsets. Of particular note are the top melt and basal growth fluxes which both attain roughly equal and opposite maxima in the autumn. The result is that while the CNRM-CERFACS models have a rather damped ice thickness seasonal cycle (Figure 2b) they display among the highest total ice growth and melt (Figure 7) because much of this happens at the same time of year, rather than being mostly compartmentalised into different seasons. Unlike for the

CMCC models, the unusual behaviour of the CNRM-CERFACS fluxes is likely reflective either of a diagnostic error or of a problem with the underlying GELATO ice thermodynamics but is not yet fully understood (David Salas & Rym Msadek, personal communication).

The mushy-layer models (NCC & NCAR) are distinguished most by exceptionally low basal conductive fluxes during the freezing season, partly because of additional unreported energy flux terms in the basal energy balance. These models are also

characterised by relatively low melt & growth fluxes, more similar to the IMBs than some other models, particularly in the case of the top melting. The lower growth flux may be partly explained by more of the ice growth being contained in the frazil ice growth term (which cannot be evaluated by IMBs), as described in Sect. 4.2 above. However, the amplitude of the ice thickness seasonal cycle in Figure 2c is also lower than for other models at similar ice thicknesses, suggesting that at least part of this is a genuine model difference. As a model group, the damped seasonal cycles are not associated with unusually

high annual mean ice thickness which would otherwise be an obvious cause (the converse of CMCC). In fact ice thickness varies greatly across these models, correlated with net radiative flux and global & Arctic temperature.

The other distinguishing feature of these models noted in Section 3 is their relatively low annual mean net SW flux, and correspondingly high net LW flux. While these could be partly related to cloud properties, they are also consistent with the relatively damped seasonal cycle of ice growth & melt in these models. It is plausible that the mushy-layer scheme, through

its action on the low basal conductive flux, may be playing a role in at least the reduced sea ice growth of these models. In Figure 4d it was seen that these models display a much higher net downward LW flux than others, indicating that these models simulate lower surface temperatures for a given downwelling LW flux. Lower conduction of heat through sea ice is a possible mechanism responsible for this.

The GSI8.1 models (CSIRO-ARCCSS, MOHC & NIMS-KMA) have among the highest annual mean ice thicknesses,

smallest LW fluxes and coldest global & Arctic temperatures. They display somewhat higher growth & top melt fluxes than the mushy-layer models. They are distinguished particularly by high summer top melt and top conduction fluxes, and correspondingly strong sensible heat gain, along with notably low ocean heat fluxes, likely associated with the lack of penetrating solar radiation. Similarly to the CMCC models, many of these may be responding in a largely physically realistic manner to a cold climate state, but the seasonal cycle of growth and melt may still be too amplified.

The remaining models (from IPSL and MRI) share no common model components or code but are strikingly similar in their simulation of basal melt & top conduction (as well as in annual mean ice thickness), both being amplified relative to the IMBs. MRI-ESM2-0 however is colder both globally and in the Arctic. It simulates less top melting in summer and basal conduction in winter, and has a less amplified seasonal cycle, all realistic effects of a colder climate. It also simulates less net SW in summer and more net LW in winter.

In summary, the fluxes of various model groups largely respond in a physically realistic manner to the differences between the base model climates, and the differences between the model thermodynamic choices, with the exception of the autumn maxima of the CNRM-CERFACS models. As a group, however, fluxes tend to be higher as a function of ice thickness than is estimated by the IMBs. This discrepancy is now evaluated in more detail.

## 5 Accounting for the IMB sampling bias

As discussed in Section 4, thin ice is likely under-sampled by the IMBs and this may introduce a bias to evaluations of fluxes that systematically vary with ice thickness (notably basal growth and the conduction fluxes). To account for this, we evaluate fluxes at each model grid point as a function of ice thickness, applying the analysis first to the top conductive flux in the Beaufort Sea region, for the months of December, January, and February. For each model, we select 100 points at random and produce scatter plots of top conductive flux against ice thickness, comparing these to monthly mean top conductive flux and ice thickness from the IMBs (Figure 8a). The same comparison is shown as a set of histograms, sampling the full sets of points for each model (Figure 8b). The sampling bias of the IMBs is demonstrated, with ice thinner than 1m barely sampled – although ice thicker than 3m is also sampled very little. The thin (thick) ice tendencies of the CMCC (GSI8.1) model groups are also demonstrated. Modelled fluxes are shown to be biased towards higher magnitudes, even for the same range of ice thicknesses. For the 1-2 m and 2-3 m histogram categories, each of which contains large numbers of IMB points, all model distributions are significantly different to the IMBs. At first sight this suggests the model bias towards high conductive fluxes is real. However, two other effects must be considered.

Firstly, ice thickness is not the only variable directly affecting conductive flux; snow depth also has a strong effect, as snow is a very powerful insulator. To account for snow depth and ice thickness simultaneously, we define the thermal insulance of the snow-ice column as

$R_{ice} = \frac{h_I}{k_I} + \frac{h_S}{k_S}$    (1)

where $h_I$, $k_I$, $h_S$ and $k_S$ represent ice thickness, ice conductivity, snow depth and snow conductivity respectively.

Secondly, the model points plotted in Figure 8 do not represent single floes (like the IMBs). Instead, they are averages over grid cells typically tens of km in width, all of which include ice of multiple thicknesses due to the models' sub-gridscale ice thickness distributions. This also biases the comparison, as a model grid cell of average ice thickness 2m will tend to allow

more conduction than a single ice column of thickness 2m. This is due to the nonlinear relationship between conduction and ice thickness, which is most sensitive for the thinnest ice. A grid cell of mean thickness 2m will likely contain, in its various

separate categories, both thinner ice which will allow much greater conduction, and thicker ice whose conduction, while lower than that of a 2m floe, is closer to this in magnitude than is that of the thinner ice. This issue is illustrated by the long negative tails on the right of Figure 8b; even grid cells of mean ice thickness 4-5m can produce substantial conductive fluxes, if they contain sufficient thin ice in their thickness distribution.

It is impossible to exactly account for this effect without knowledge of conduction flux per ice category, which is not provided by any model in the CMIP6 subset. However, the effect can be roughly estimated using category ice concentration, snow depth and ice thickness data, which is provided by six models: MOHC UKESM1-0-LL, MRI-ESM2-0, IPSL-CM6A-LR, IPSL-CM6A-LR-INCA, NorESM2-LM and NorESM2-MM. For each grid cell in the comparison, the conductance of the ice in each category is estimated from the ice thickness and snow depth. The individual category conductances are then summed together, weighted by category ice concentration, to obtain the total grid cell conductance. An 'effective' thermal insulance is then obtained from this, which represents the ability of the ice in the grid cell to conduct energy in a manner comparable to the IMB point measurements.

For the 6 models, top conductive flux is plotted both against approximate thermal insulance (derived from gridbox mean ice thickness and snow depth; Figure 9a), and against corrected thermal insulance, derived from ice thickness distribution information as described above (Figure 9b). Using the corrected thermal insulance has the effect of pushing the model points to the left, by reducing the insulance. Hence each insulance class is composed of points with thicker ice, and therefore weaker conduction, than is the case with the uncorrected insulance. This reduces the model bias further but does not eliminate it: for all ranges of thermal insulance for which substantial numbers of IMB points exist, all 6 models are still biased significantly negative relative to the IMBs. For the 1-1.5 K $m^2$ $W^{-1}$ insulance range, the IMBs measure an average top conductive flux of -18.8±4.3 W $m^{-2}$, whereas the six models simulate an average conductive flux of -29.5 ± 5.4 W $m^{-2}$.

The analysis is repeated for the North Pole region (not shown). Although top conductive flux is biased less strongly with respect to the IMBs in this region, mapping the distributions to thermal insulance classes again has the effect of reducing but not eliminating the bias. In the 1.5-2 K $m^2$ $W^{-1}$ range for example, the IMBs measure an average top conductive flux of -13.0±3.0 W $m^{-2}$, whereas the six models simulate an average of -22.5±4.2 W $m^{-2}$. All models are significantly biased low with respect to the IMBs in this class, with NorESM2-MM and the IPSL models also significantly biased low in the 1.0-1.5 K $m^2$ $W^{-1}$ class.

The top conductive flux expresses the thermal forcing of the atmosphere on the ice, but it is the basal conductive flux that modulates how this forcing drives ice growth. The basal conductive flux is biased negative less strongly than the top conductive flux, and for the mushy-layer models is biased positive. The IPSL and MOHC models display small negative biases as a function of GBM thermal insulance that are mostly removed when they are evaluated as a function of corrected insulance. MRI-ESM2-0 displays a larger bias that is not removed. The NCC models remain strongly biased positive (less conduction) for each insulance class.

The above analysis is for December-February; we also compare conductive fluxes for October-November, when ice growth is typically strongest as ice is thinner and insulance lower. As for the winter months, conductive flux bias is stronger at the

top than at the base of the ice, and greater in the Beaufort Sea than in the North Pole region. However, a larger number of IMB measurements are available for the smallest insulance classes during the autumn. In the 0.5-1 K m$^2$ W$^{-1}$ insulance class, both top and basal conductive fluxes are significantly biased low in both regions for all models, with the exception of NCC basal conductive fluxes which remain biased high.

We consider also the relationship between ice thickness and melting fluxes. It was seen in Figure 7 that the relationship between seasonal average top melting and ice thickness was rather weak. To illustrate this further, we compare July top melting fluxes against ice thickness by gridpoint for the Beaufort Sea region (not shown). This shows the cross-model relationship between ice thickness and top melt to be much weaker than the winter relationship between ice thickness and conductive fluxes, with wide distributions of top melting fluxes simulated at all thicknesses, albeit with more extremely high

values at low thicknesses. Inter-model variability dominates variability due to ice thickness, with e.g. the CMCC and CNRM-CERFACS models displaying two nearly distinct clusters at similar ice thicknesses. This could be partly because the relationship between top melting and ice thickness operates on somewhat larger scales than individual grid cells, via ice area and the ice albedo feedback.

The IMBs, too, do not display a clear relationship between ice thickness and top melt flux. In fact, there is a notable lack of

high top melt fluxes for low ice thicknesses in the IMB measurements. This may point again to the sampling problem inherent in the Lagrangian nature of the IMBs; thin ice floes subject to high melting fluxes will not survive long enough to report a flux to the IMB dataset. To address this problem, we compute daily top melting fluxes for the IMB dataset and compare these to the monthly mean modelled fluxes (Figure 10). While model grid-box mean fluxes account for the fast melting rates of thin ice irrespective of meaning period, it is necessary to refine the IMB fluxes to a finer temporal resolution

to detect these.

The 0-1 m range is still very poorly sampled by the daily IMB fluxes; examination shows that this is because it is rare for IMBs to report data right up until the point of melting out. Moreover, on the occasions this actually happens the final destruction of the floe is due to basal rather than top melting, which provides additional evidence for the relationship between ice thickness and top melting being rather weak. However the 1-2 m category provides plenty of datapoints. With

the exception of the CNRM-CERFACS models which do not report maximum top melt until September, all models have monthly grid-box mean fluxes much higher than the daily fluxes shown by the IMBs in this category.

## 6 Conclusions

We have evaluated ice energy fluxes for a range of CMIP6 models using a dataset of equivalent fluxes derived from IMBs. In most cases, the fluxes vary between the models in ways consistent with the diverse model climates, and underlying ice

thermodynamics schemes. Collectively however, the models tend to simulate greater melt, growth and conduction fluxes than are measured by the IMBs for similar ice thicknesses.

In more detail, the model states in the evaluated period 1985-2014 vary from warm, with thin sea ice (the CMCC models) to cold, with thick sea ice (particularly some of the GSI8.1 models). Models with thicker sea ice tend to simulate smaller melt, growth & conduction fluxes, and vice versa, which is consistent with ice thermodynamics theory. Model clusters are also influenced by choices in ice thermodynamics parameterisation. For example, the GSI8.1 models tend to model higher top melting relative to ice thickness than many others, and the ice thickness seasonal cycles are relatively more amplified, and further from the IMB values. Conversely, the mushy-layer models tend to simulate lower basal conductive fluxes than others, and their seasonal cycles are relatively more damped, and closer to the IMB values. The flux simulation differences of these model clusters are likely to be linked to the lack of penetrating solar radiation, and to the inclusion of mushy-layer thermodynamics, respectively. Hence while the atmosphere exerts a first-order control on the sea ice state, the sea ice parameterisation choice has a second-order effect on how growth and melt respond to this.

A large part of the discrepancy between IMBs and models is influenced by the ice thickness sampling bias, the tendency for IMBs to be placed in thicker, more robust ice floes, and by the Lagrangian-Eulerian sampling bias, the tendency for thin ice floes to measure faster growth and melt fluxes. Despite this, the ice thickness distribution analysis provides strong evidence that a portion of the conduction/growth bias is real. Additional circumstantial evidence has been provided that a portion of the melting bias is also real, by comparing modelled fluxes to IMB-measured daily fluxes.

We note that as a purely thermodynamic analysis, this study does not attempt to give any account of the influence of differing ice dynamics on modelled ice volume, either via dynamic atmospheric forcing or ice dynamic scheme choices. While this factor is likely of secondary importance to the atmospheric thermal forcing, its influence on modelled ice volume may be similar to that of ice thermodynamic choices. For example, Keen et al. (2021) found that ice advection accounted for between 9% and 30% of the total annual sea ice loss in CMIP6 models.

The evaluation underlines the value of the IMB observations in aiding detailed analysis of sea ice process modelling. Future deployments of these instruments would be very useful to study changes in ice thermodynamics as sea ice continues to become thinner and less extensive.

The evaluation also suggests that there is value in continuing to improve the accuracy of sea ice thermodynamics simulation. Differences in ice thermodynamics choices are shown to have a clear, measurable effect on the ice melt, growth and conduction fluxes that are simulated for particular ice thicknesses that is almost certainly reflected in how the sea ice state responds to a given atmospheric forcing. This is particularly the case for the penetrating solar radiation inclusion and for the mushy-layer parameterisation, although the latter may suppress conduction too much owing to the issues identified by Plante et al. (2024).

The study also proves the importance of providing detailed energy budget information to the CMIP6 archive. Although the CMIP6 subset contained reasonable model diversity, many additional models were of complexity sufficient to justify submitting the energy flux diagnostics to the CMIP6 archive, and it would be of great value if a wider variety of models provided these for future CMIP iterations.

## Code availability

The code used to analyse the IMB data is published in two repositories, corresponding to two stages of the analysis. The code used to read, quality control and process the data into consistent quantities on consistent time points can be downloaded from https://doi.org/10.5281/zenodo.3975692 (West, 2020a). The code used to produce datasets of monthly mean energy fluxes from this processed data, as well as that used to produce the daily data used in Figure 10, can be downloaded from https://doi.org/10.5281/zenodo.3971736 (West, 2020b).

The code used to analyse model data, and calculate timeseries, multiannual means and ice-area weighted statistics, and the code used to produce Figures 1-10, is published at https://doi.org/10.5281/zenodo.12518762 (West, 2024).

## Data availability

The raw IMB data are publicly available and can be downloaded from http://imb-crrel-dartmouth.org/results/ (last access: 20 April 2020, Perovich et al., 2020). The processed IMB data and the derived dataset of monthly mean fluxes are published at https://doi.org/10.5281/zenodo.3773811 (West, 2020c) and https://doi.org/10.5281/zenodo.3773997 (West, 2020d) respectively. The derived dataset of daily mean fluxes used in Figure 10 is also published at https://doi.org/10.5281/zenodo.3773997 (West, 2024d).

The diagnostics from CMIP6 used in this study are all publicly available and can be downloaded from the CMIP6 archive at https://esgf-index1.ceda.ac.uk/projects/cmip6-ceda/.

## Author contribution

The analysis was devised, undertaken and documented by AW. EB gave feedback at each stage of the analysis and on the final paper drafts.

## Competing interests

The authors declare that they have no conflict of interest.

## Acknowledgements

This study was supported by the Met Office Hadley Centre Climate Programme funded by DSIT. The authors thank Emma Fiedler and Jeff Ridley for helpful suggestions concerning reference datasets.

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

| Grouping | Institute | Model | Atmosphere | Ocean | Sea ice | Thickness category bounds (m) | Vertical thermodynamic layers (ice+snow) | Salinity treatment | Melt pond parameterisation | Penetrating solar radiation |
|---|---|---|---|---|---|---|---|---|---|---|
| GSI8.1 | CSIRO-ARCCSS | ACCESS-CM2 | UM v10.6 (GA7.1) | GFDL MOM5 | CICE5.1.2 (GSI8.1) | 0.6, 1.4, 2.4, 3.6 | 4I+1S | Prescribed | Topographic | N |
| | MOHC | HadGEM3-GC31-LL | | NEMO3.6 | | | | | | |
| | MOHC | HadGEM3-GC31-MM | | NEMO3.6 | | | | | | |
| | MOHC | UKESM1-0-LL | | NEMO3.6 | | | | | | |
| | NIMS-KMA | UKESM1-0-LL | | NEMO3.6 | | | | | | |
| Mushy-layer | NCAR | CESM2 | CAM6 | POP2 | CICE5.1.2 | 0.645, 1.391, 2.470, 4.563, 9.338 | 8I+3S | Fully prognostic (mushy-layer) | Level-ice (H13) | Y |
| | NCAR | CESM2-WACCM | WACCM6 | POP2 | | 0.645, 1.391, 2.470, 4.563, 9.338 | | | | |
| | NCC | NorESM2-LM | CAM6-Nor | BLOM | | 0.6, 1.4, 2.4, 3.6 | | | | |
| | NCC | NorESM2-MM | CAM6-Nor | BLOM | | 0.6, 1.4, 2.4, 3.6 | | | | |
| CMCC | As grouping | CMCC-CM2 | CAM5 | NEMO3.6 | CICE4 | 0.64, 1.39, 2.47, 4.57 | 4I+1S | Prescribed | Level-ice (H12) | Y |
| | | CMCC-ESM2 | | | | | | | Level-ice (H12) | |
| IPSL | As grouping | IPSL-CM6A-LR | LMDZ6A-LR | NEMO3.6 | LIM3.6 | From formula | 2I+1S | Prognostic bulk salinity | Parameterised via albedo | Y |
| | | IPSL-CM6A-LR-INCA | | | | | | | | |
| MRI | As grouping | MRI-ESM2 | MRI-AGCM3.5 | MRI-COMv4 | In-house (within ocean model) | 0.6, 1.4, 2.4, 3.6 | 1I+1S | Prescribed | Parameterised via albedo | Y |
| CNRM-CERFACS | As grouping | CNRM-CM6-1 | ARPEGE-Climat v6.3 | NEMO3.6 | GELATO6 | 0.3, 0.7, 1.2, 2 | 9I+1S | Prognostic bulk salinity | Parameterised via albedo | Y |
| | | CNRM-CM6-1-HR | | | | | | | | |
| | | CNRM-ESM2-1 | | | | | | | | |

*Table 1. The CMIP6 subset of models, their components and characteristics of thermodynamic parameterisations. In the melt pond column, H12 and H13 refer to Holland et al. (2012) and Hunke et al. (2013) respectively.*

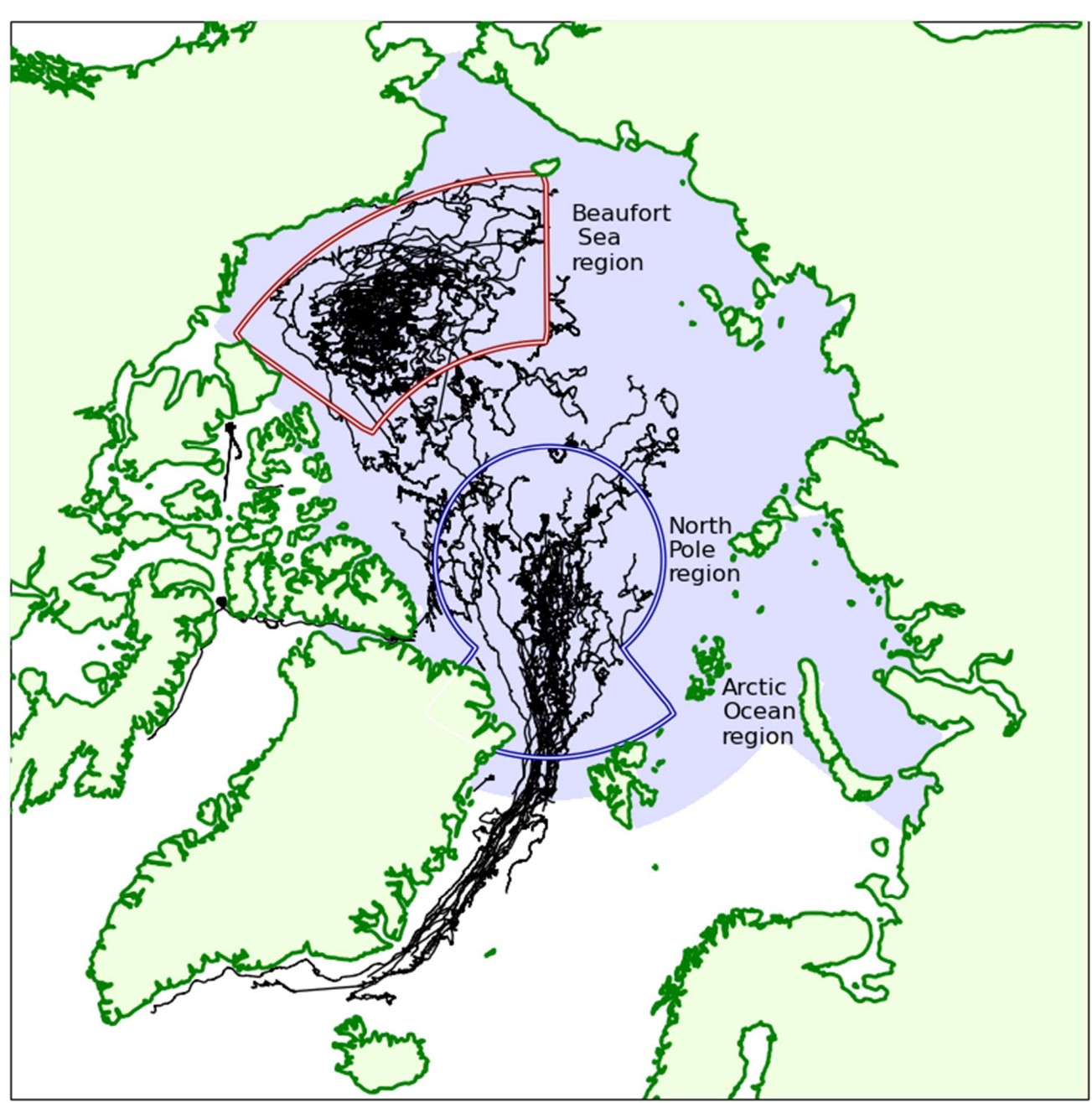

*Figure 1. A map of all IMB tracks in the Arctic between 1993-2015 used in this study. The Arctic Ocean region (blue shading), and the North Pole (dark blue box) and Beaufort Sea (dark red box) subregions used in the analysis are indicated.*

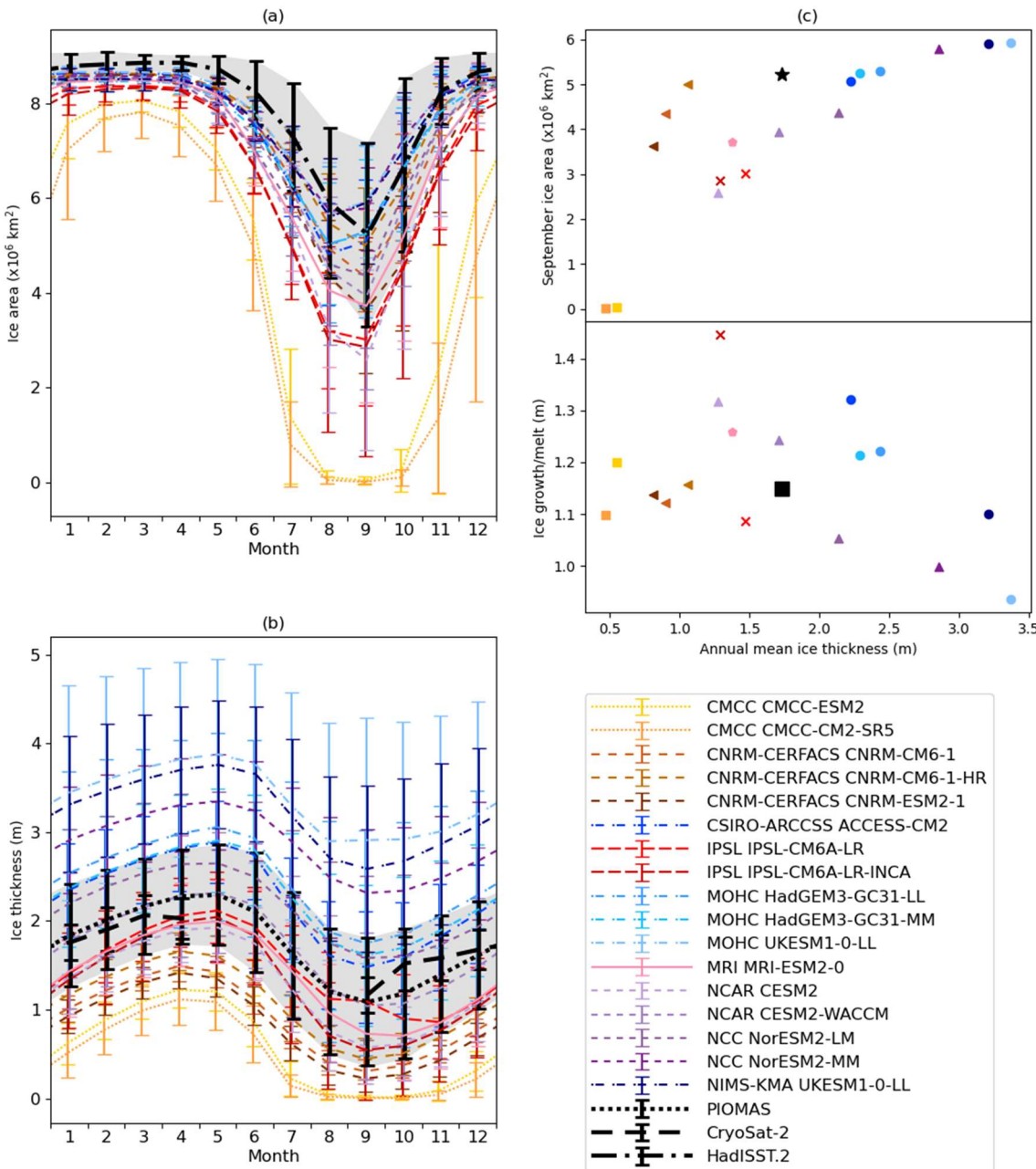

*Figure 2. (a) Total Arctic Ocean ice area, and (b) average Arctic Ocean ice thickness in the CMIP6 subset (1985-2014 average with bars denoting twice the interannual standard deviation). HadISST1.2 for area, and PIOMAS and CryoSat-2 for ice thickness, are shown for comparison; for CryoSat-2, the period shown is 2011-2020 but for the other datasets 1985-2014 is used. (c) Scatter plot of annual mean ice thickness against September ice area (top panel) and ice growth/melt diagnosed from October-September mean minus April-May mean ice thickness (bottom panel).*

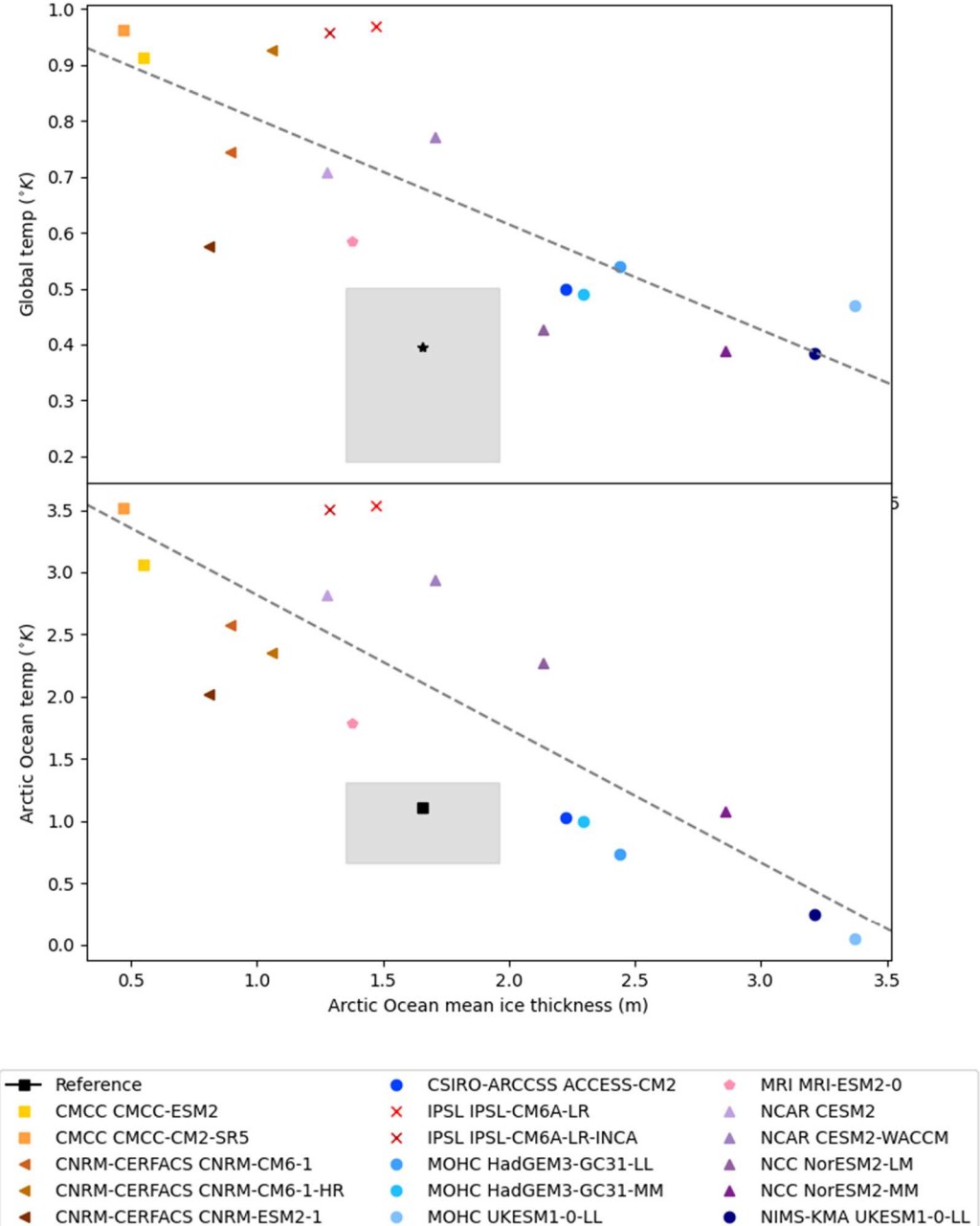

Figure 3. 1985-2014 Arctic Ocean region mean ice thickness compared to global (top) and 2m air temperature anomaly over the Arctic Ocean region relative to 1850-1899 (bottom). The black symbols and grey filled regions represent observational estimates and uncertainty intervals for ice thickness and temperature anomaly derived as described in Section 2.3.


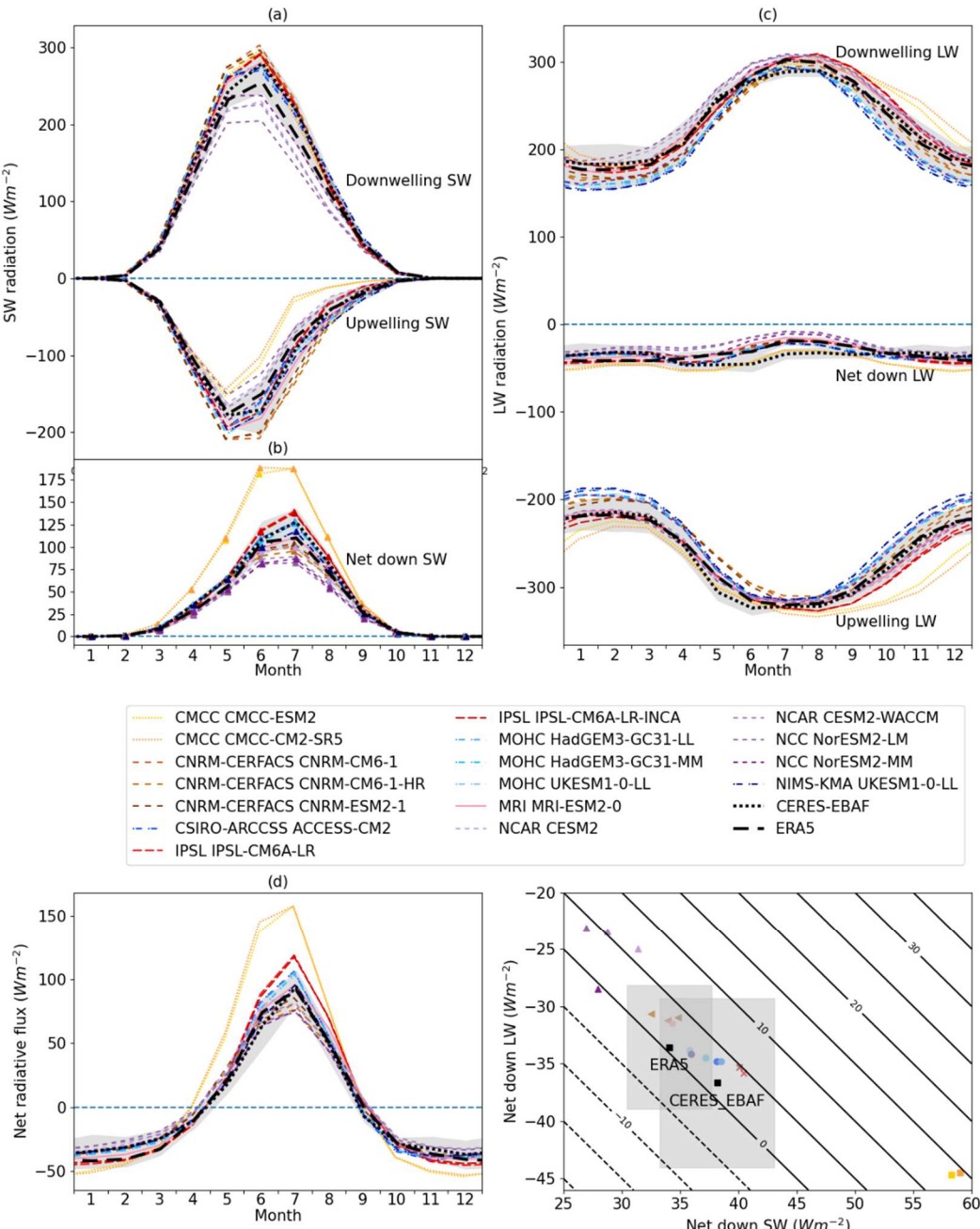


*Figure 4. Evaluation of radiative fluxes in the CMIP6 subset relative to ERA5 reanalysis and CERES-EBAF satellite dataset. (a) from top to bottom, downwelling, upwelling and net down SW radiation; (b) from top to bottom, downwelling, net down and upwelling LW radiation; (c) total net radiative flux; (d) scatter plot of annual mean net SW versus annual mean net LW, with isolines of total net radiative flux overplotted. For (a), net down SW radiation is distinguished by triangle markers*

*where model spread overlaps with downwelling SW.*

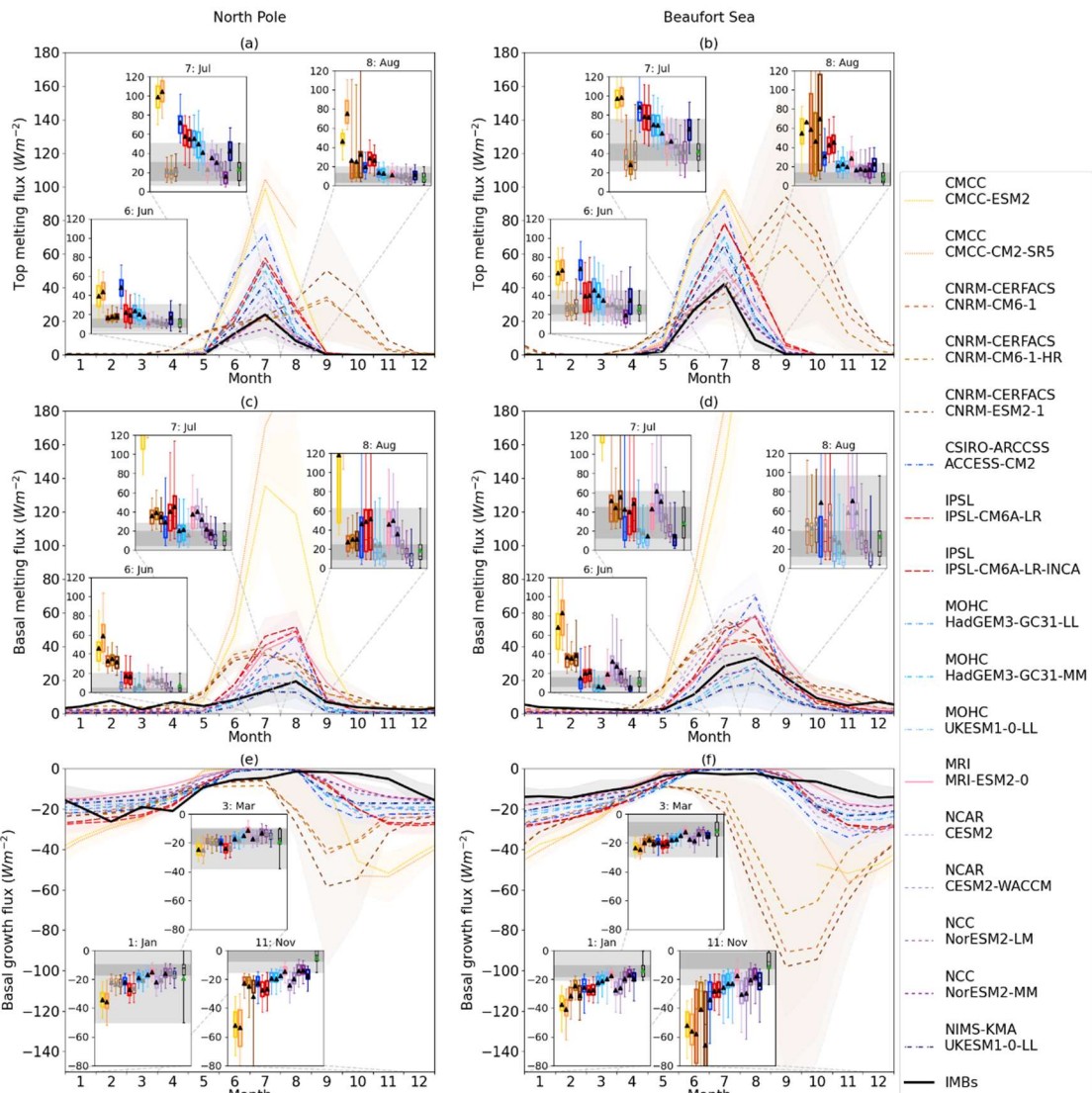

*Figure 5. Seasonal cycles of (a,b) top melting flux; (c,d) basal melting flux; (e,f) basal growth flux in North Pole (left column; a,c,e) and Beaufort Sea (right column; b,d,f) regions, with ice area-weighted mean and standard deviation shown.*
*Means and standard deviations are taken across all grid cells in the respective regions and across all years in the study period (1985-2014). For each flux and region, inset plots show boxplots of ice area-weighted statistics across all grid cells for three key months of the year: June-August for melting fluxes; November, January & March for growth fluxes. IMB distribution mean & standard deviation, and boxplots (black box and filled area), are shown for each flux and region for comparison. Modelled distributions judged significantly different to the IMBs are distinguished by bold outlines and black*
*mean triangles.*

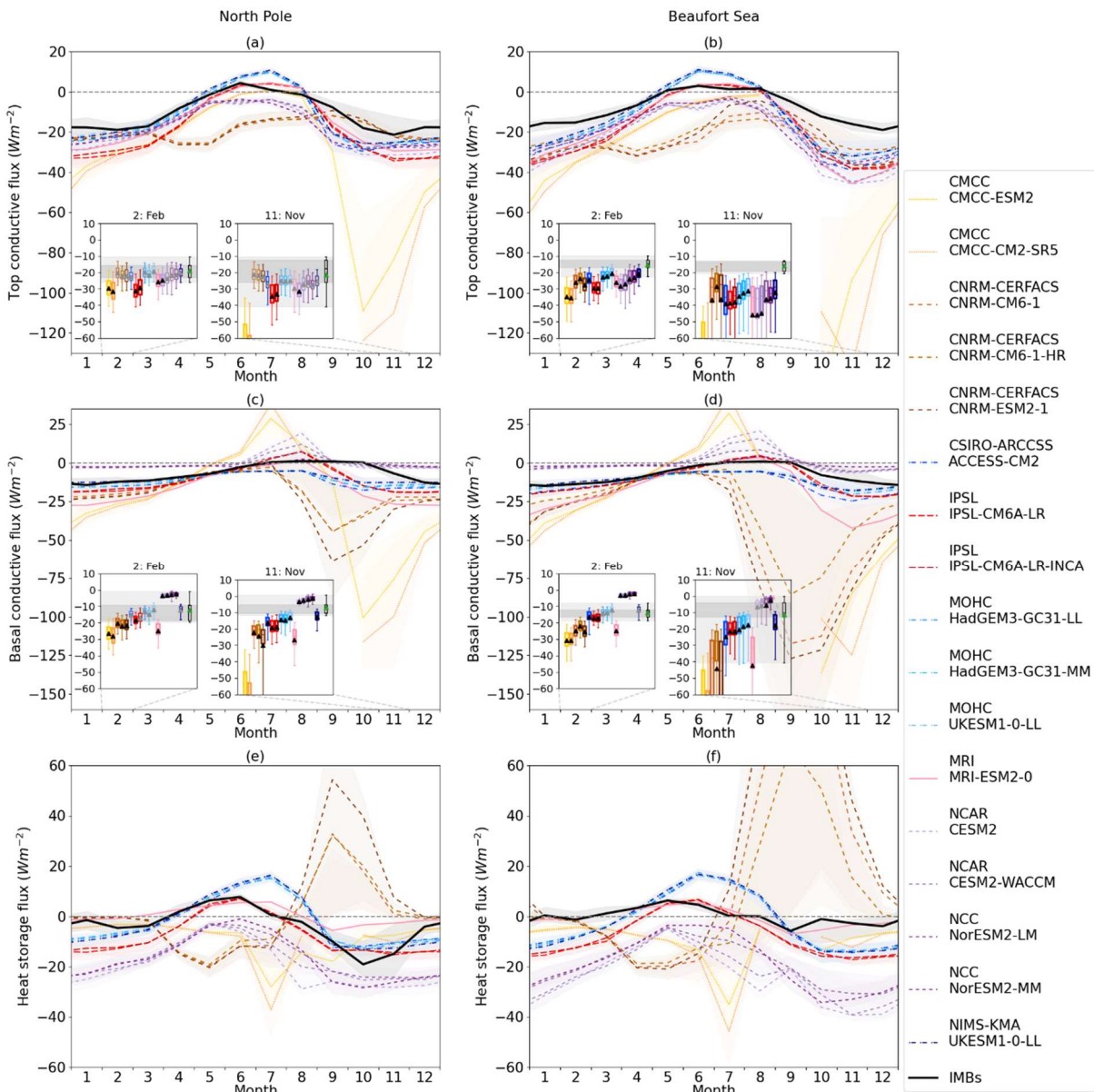

*Figure 6. Seasonal cycles of (a,b) top conduction flux; (c,d) basal conduction flux; (e,f) heat storage flux for the North Pole (left column; a,c,e) and Beaufort Sea (right column; b,d,f) regions, with ice area weighted mean and standard deviation shown. Means and standard deviations are taken across all grid cells in the respective regions and across all years in the study period (1985-2014). For the conduction fluxes in each region, inset plots show boxplots of ice area-weighted statistics across all grid cells for January and November. IMB distribution mean & standard deviation, and boxplots (black box and filled area), are shown for each flux and region for comparison.*

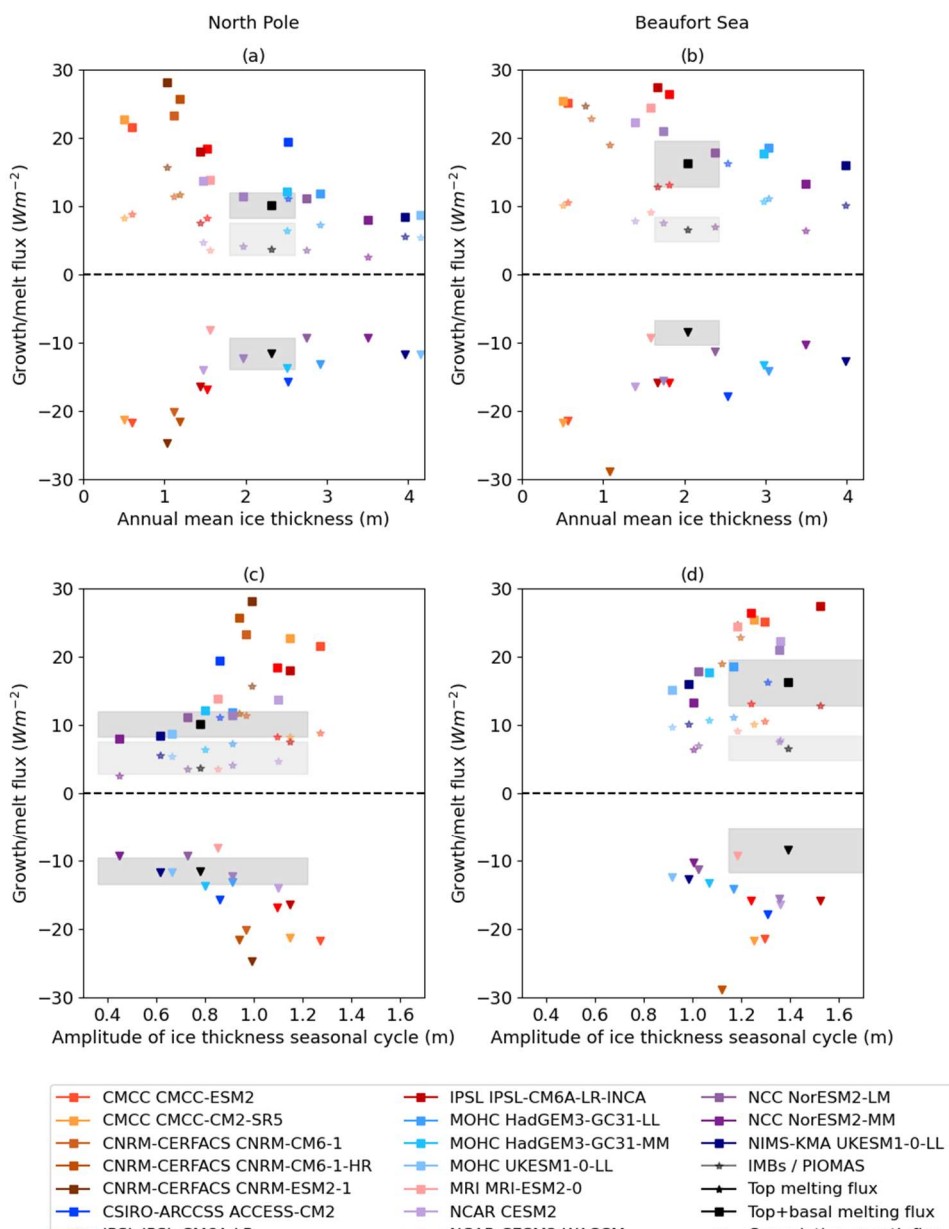

*Figure 7. Seasonally averaged melt & growth fluxes (weighted by ice area for each month and region) plotted against annual mean ice thickness (a,b; top row) and ice thickness seasonal cycle amplitude (c,d; bottom row), for North Pole (a,c) and Beaufort Sea (b,d) regions. Showing total melt (squares), top melt (stars) and total growth (triangles), as defined in the text. Shaded regions represent observational uncertainty: for melt and growth fluxes, IMB-derived uncertainty as diagnosed in Section 3.3 of West et al. (2020); for ice thickness, uncertainty derived from the differences between PIOMAS and CryoSat-2 as described in Section 2.3.*

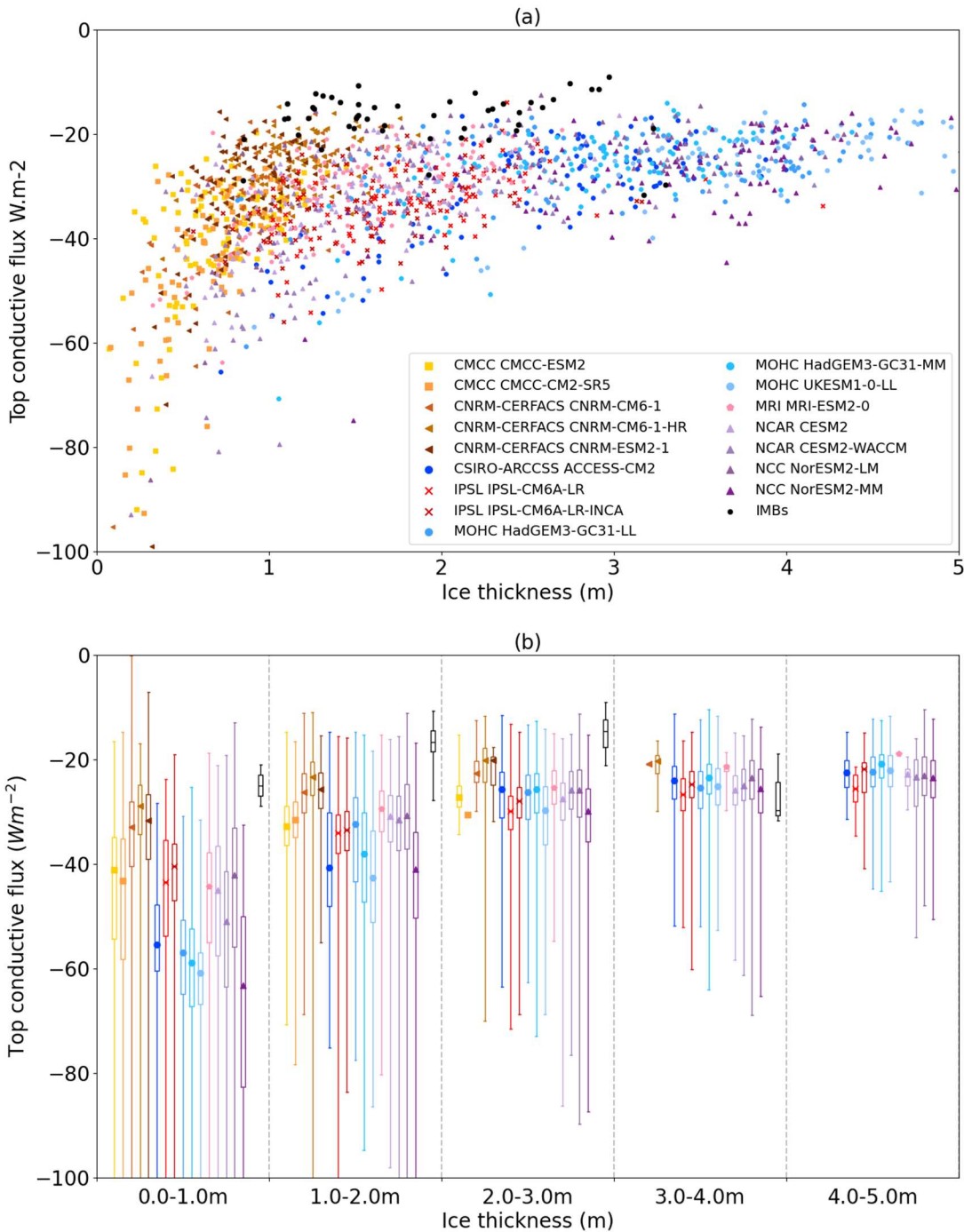

*Figure 8. (a) Scatter plot of top conductive flux against ice thickness for models and IMBs; for clarity, a random set of 100 points is selected for each model. (b) The same comparison, shown as a series of box plots, but with all model points sampled. Note that the 0-1 m and 3-4 m classes contain only 2 and 3 IMB points respectively.*

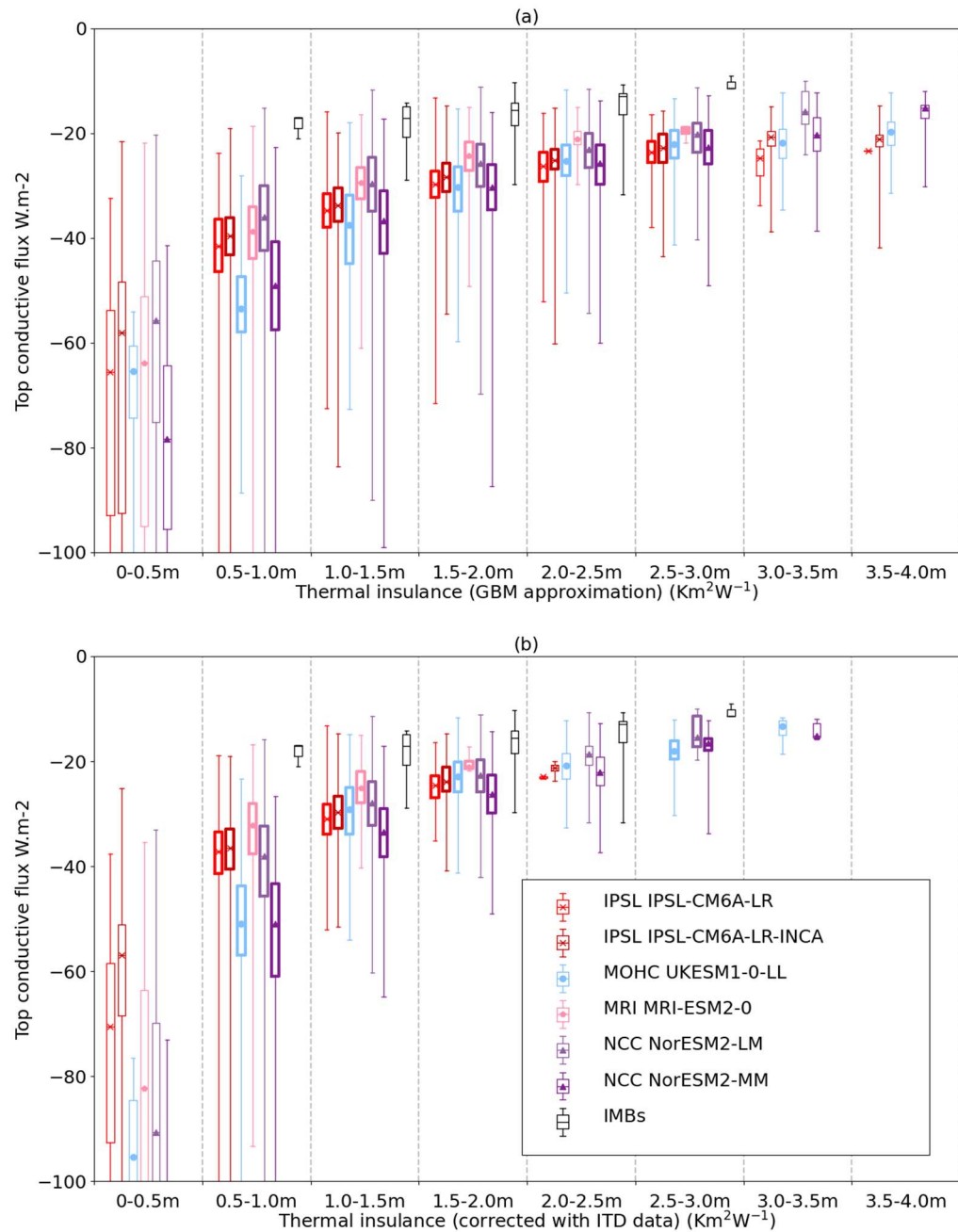


*Figure 9. (a) Top conductive flux boxplots as a function of thermal insulance, calculated from model gridbox mean ice thickness and snow depth. (b) Top conductive flux boxplots as a function of thermal insulance calculated from sub-grid ice thickness distribution properties. For each model and insulance class, distributions significantly different to the IMBs are highlighted in bold.*

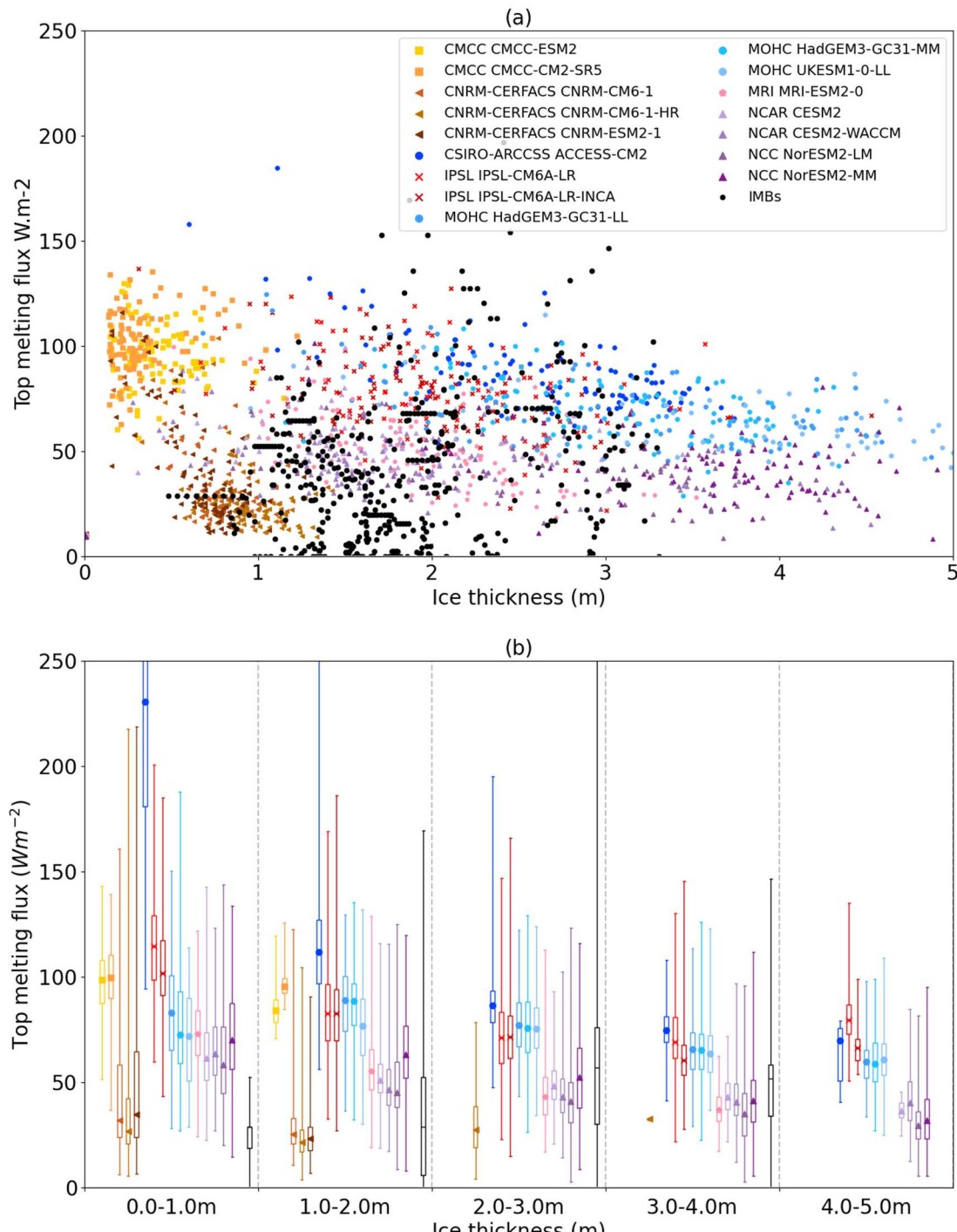


*Figure 10. (a) Scatter plot of monthly mean top melting fluxes compared to daily mean IMB fluxes, for a randomly selected subset of the Beaufort Sea region. (b) Bar plot for the same comparison, using the full model distributions.*