# Peer review of "CMIP6 models overestimate sea ice melt, growth & conduction relative to ice mass balance buoy estimates"

_Geoscientific Model Development, 2024_

## Referee Comment (RC2)

Review of:

*"CMIP6 models overestimate sea ice melt, growth & conduction relative to ice mass balance buoy estimates"* by Alex E. West and Edward W. Blockley.

This manuscript discusses the performance of CMIP6 models in reproducing the sea ice thermodynamics as measured from Ice Mass Balance (IMB) buoys in the central Arctic and the Beaufort Sea. In particular, the authors inter-compare the simulated sea ice growth and melt from different CMIP6 members and discuss differences in terms of the simulated sea ice state (yearly mean thickness and area). They effectively group the selected CMIP6 members according to their component characteristics, allowing them to identify common patterns and the source of discrepancies. They find that in general, the simulated thermodynamic fluxes in the selected CMIP6 members respond in a realistic manner to the simulated climatological sea ice state but overestimate the magnitude of these fluxes.

I find that the manuscript is very interesting, pertinent and of good quality. The presented results are insightful on model sensitivities, and on the importance of an accurate representation of the heat fluxes at the air-ice and ice-ocean interfaces. The use of in-situ observations to evaluate the internal sea ice thermodynamics is also a significant contribution.  This analysis relevant for publication in the Journal of Model Developments.

Nonetheless, I find that the more general contributions in the manuscript are sometimes difficult to isolate within the detailed results. There are also a few needed corrections and clarifications to better guide the reader towards the main conclusions.

I thus recommend this manuscript to be accepted for publication, after major revisions, as listed below.

Mathieu Plante

General comments:

- There is a tendency to use overly concise wordings, at the cost of preciseness and sometimes accuracy. This is especially present in introduction, where some steps in the reasonings are skipped, likely because they seem obvious to the authors, but effectively leaves it to the reader to work it out. I believe it is worth spending more wordings to be more precisions, especially for processes that are later referred to in the analysis.

- I would like to have more information in the method sections on the metrics used: e.g., which are diagnostics (i.e., growth and melt rate terms) vs. derived from the internal temperature profiles (conductive fluxes), and how they relate to the method used to get the in-situ values from IMBs. This would also help to shift the focus on the use of IMBs to assess CMIP6 models (for a GMD manuscript).

- I find it difficult to fully discuss ice top heat conduction without discussing the snow layer. This is somewhat covered in section 5, but could be mentioned earlier and throughout the analysis.

- Discussing the mushy layer model results, the authors vaguely refer to un-reported terms in the basal heat balance. This should be more more specific, otherwise I find it somewhat difficult to interpret these model results. I suspect that the authors are referring to the treatment of sea ice congelation not fully accounting for the conductive heat flux in the CICE mushy layer congelation scheme (see Plante et al., 2024), but I am not sure. If it is so, then it is important adapt the discussion accordingly: it is not a "missing diagnostic term", but some flux sent to the ocean during congelation. As it is congelation-related, it could explain the lower growth flux but should not directly affect the melt flux.

- I find it a bit confusing that the selected CMIP6 members are referred to as the "IMB sample", given that they are compared to IMB buoys.

- Figures are often mis-referenced in the text.

Specific points and edits suggestions (some repeating the points above):
- L24: "*although some improvement in agreement with reference datasets with model resolution and model complexity is discernible*": This wording is not clear, please be more specific.

- L26-32: This paragraph is confusing, mainly because it is so concise that it becomes too vague. It would be worth expanding on these processes so that they are well understood by the reader before getting into the analysis. For instance, thicker ice melts more than thin ice: because of the larger area of ice surviving longer through summer?

- L30: it is not clear if "both" refers to the growth and melt, or to the processes.

- L34: This sentence is confusing: it is not clear which climate variables you are referring to, and how it relates to the complexity of sea ice volume processes.

- L39: "Although evaluation of the internal processes of the sea ice is in principle even more difficult": This is a bit subjective, unless there is a reasoning added to this statement.

- L42-45: This is also tedious to follow for readers not familiar with this study. I think it is worth adding some precisions.

- L65: why not simply "CMIP6 subset" instead of IMB? It would be less confusing when discussing models vs IMB observations.

- L68: fast ice growth -> rapid ice growth (to avoid confusion with fast ice)

- L77: You could cite Bitz and Lipscomb (1999) here.

- Section 2.2.: Are all these IMBs all from CRREL with 10cm vertical resolution, or is there a number of them with higher resolution (e.g., SAMS, SIMB3, etc.)? It would be useful to indicate how they relate/compare to other data (for instance MOSAIC IMBs, etc., e.g. Koo et al., 2020)

- L116: "was found to demonstrate well" -> displayed? Was used to characterize?
  It currently sounds like the quality of the observations were assessed against other unnamed references.

- L121: Please add reference.

- L125: Do you mean that you use both the sounders elevation measurements and the temperature profiles to determine the material interface positions? Also, I would like to have a measure of the uncertainties, and how it compares with other methods (e.g., see Richter et al., 2023)

- Section 2.3: Some of these data are also based on models, which are somewhat related to the components in some of the CMIP6 members. Could this interfere with the results? For instance, PIOMAS is, to my understanding, based on the POP model, which is also used in some of the CMIP6 members.

- L143: "the sea ice state simulation of the IMB subset": rephrase. Perhaps: the sea ice state simulated by the subset models?

- L144: "we restrict *the* evaluation"

- L155-156: This could be presented with respect to the PIOMAS and CryoSat-2 seasonal cycle.

- L170 suggestion: "There is strong correlation (0.81) between […]"

- L172-178: This paragraph is a bit difficult to follow. Some revisions would be helpful. E.g.:
    o "We compare *the* annual mean ice thickness to *the anomaly in global 2m air temperature relative to the 1850-1899 average*".

       o   Clarify "Arctic Ocean temperature": sounds like ocean temperature. I believe that you rather refer the T2m.

- L197: "Figure 2" -> I believe you are referring to Figure 1. Many other figures are also mis-referenced (e.g. Figure 7,8,9)

- Figure 4: Remove the "down" from the labelling of the SW down as it shows upwelling SW. It is also difficult to identify which curves is the net or downwelling radiation as they are not completely staggered (due to the CMCC curves). I recommend moving the net SW in a separate plot.

- Figure 5: Missing information in the caption. The box plots are distributing the monthly means… for each year in the study period?

- L246: typo ::  or -> for

- L263-267 (and also L501-502): I am not sure I understand the 2$^{nd}$. Are you referring to the fact that IMBs will not sample the new ice forming in leads that form during the observation period?

- L290: "but display negative conduction in summer". This is interesting. Is this computed from the ice interior, or the conductive flux diagnostic? Is this indicating that the surface temperature is colder than the ice interior despite warmer air temperature?

- L310-311 (also L328-330): Please clarify : I do not get how a missing diagnostic term in the energy balance would impact the conduction computed from the simulated temperature profiles. However, as I mentioned above, I believe that you may be referring to the mistreatment of the energy balance in the mushy layer congelation scheme (not only a diagnostic), which results in a wetter and warmer ice base.

- L344-345: In the CICE mushy layer scheme, a significant portion of the ice growth happens via frazil formation (DuVivier et al., 2022), again due to the treatment of the conductive flux in the congelation scheme. The fact that the frazil flux is not included here may thus impact the mushy group more than the other models.

- Figure 7: In the panel c, the IMB points lie outside of the IMB uncertainty shading… Is that a plotting error?

- L350: Is there more behind this attribution? It is not obvious when looking at Fig. 7 (which is also is referred to as Fig. 8 in the text). One could argue that the relationship is weak even among the model groups (e.g., there is low top melting in the purple models, without much of a slope).

- L358: "overlapping between ice growth and melt terms" -> rephrase: the terms are not overlapping, their season is.

- L375 (comment): I think that the fact that the CNRM-CERFACS models also display large conduction indicates that we have here a thermodynamic issue, rather than a diagnostic one (i.e., a problem in diagnostics would not influence the simulated internal temperature profiles).

- L388: It is also likely that lower growth is partly associated to the missing frazil contribution, which is more impactful using the current CICE mushy layer scheme.

- L389-391: I am not sure I got this right, perhaps it is worth spending more words here to clarify. I.e, the high (outward?) net LW flux is indicative of a cold surface temperature, despite the warmer atmosphere?

- L440: "pushing points to the left" -> towards smaller insulance ?

- Figure 9: It looks like there is also vertical differences in the distributions. Are the conductive fluxes also computed differently in panel a and b?

- L466-475: I feel like there is a missing point here: to me, this analysis is indicating that the relationship between ice thickness and top-melt is a large scale one (i.e., it related to the ice Area, via the albedo effect), and thus is not showing when looking at individual grid points and IMBs.

- L476: ameliorate -> address?

References:

Bitz, C. M., and W. H. Lipscomb (1999), An energy-conserving thermodynamic model of sea ice, *J. Geophys. Res.*, 104(C7), 15669–15677, doi:10.1029/1999JC900100.

Koo, Y., Lei, R., Cheng, Y., Cheng, B., Xie, H., Hoppmann, M., Kurtz, N.T., Ackley, S.F., Mestas-Nunez, ˜ A.M., 2021. Estimation of thermodynamic and dynamic contributions to sea ice growth in the Central Arctic using ICESat-2 and MOSAiC SIMBA buoy data. Remote Sens. Environ. 267, 112730. https://doi.org/10.1016/j.rse.2021.112730.

Plante, M., Lemieux, J.-F., Tremblay, L.B., Tivy, A., Angnatok, J., Roy, F., Smith, G., Dupont, F., (2024), Using Icepack to reproduce Ice Mass Balance buoy observations in landfast ice: improvements from the mushy layer thermodynamics, The Cryosphere, 18, 1685–1708, https://doi.org/10.5194/tc-18-1685-2024.

Richter, M. E., Leonard, G. H., Smith, I. J., Langhorne, P. J., Mahoney, A. R., and Parry, M.: Accuracy and precision when deriving sea-ice thickness from thermistor strings: a comparison of methods, J. Glaciol., 69, 879–898, https://doi.org/10.1017/jog.2022.108, 2023.

---

## Author Comment (AC2)

**Response to second review of 'CMIP6 models overestimate melt, growth & conduction fluxes relative to ice mass balance buoy estimates' (Mathieu Plante)**

Throughout this response, the original review is reproduced in black; our responses are shown in red. References are given at the end.

Review of: "CMIP6 models overestimate sea ice melt, growth & conduction relative to ice mass balance buoy estimates" by Alex E. West and Edward W. Blockley.

This manuscript discusses the performance of CMIP6 models in reproducing the sea ice thermodynamics as measured from Ice Mass Balance (IMB) buoys in the central Arctic and the Beaufort Sea. In particular, the authors inter-compare the simulated sea ice growth and melt from different CMIP6 members and discuss differences in terms of the simulated sea ice state (yearly mean thickness and area). They effectively group the selected CMIP6 members according to their component characteristics, allowing them to identify common patterns and the source of discrepancies. They find that in general, the simulated thermodynamic fluxes in the selected CMIP6 members respond in a realistic manner to the simulated climatological sea ice state but overestimate the magnitude of these fluxes.

I find that the manuscript is very interesting, pertinent and of good quality. The presented results are insightful on model sensitivities, and on the importance of an accurate representation of the heat fluxes at the air-ice and ice-ocean interfaces. The use of in-situ observations to evaluate the internal sea ice thermodynamics is also a significant contribution. This analysis relevant for publication in the Journal of Model Developments.

Nonetheless, I find that the more general contributions in the manuscript are sometimes difficult to isolate within the detailed results. There are also a few needed corrections and clarifications to better guide the reader towards the main conclusions.

I thus recommend this manuscript to be accepted for publication, after major revisions, as listed below.

We thank the reviewer for their positive and constructive remarks.

Mathieu Plante

General comments:

- There is a tendency to use overly concise wordings, at the cost of preciseness and sometimes accuracy. This is especially present in introduction, where some steps in the reasonings are skipped, likely because they seem obvious to the authors, but effectively leaves it to the reader to work it out. I believe it is worth spending more wordings to be more precisions, especially for processes that are later referred to in the analysis.

We understand the point the reviewer is making. We think there is scope for a fuller discussion of the relationship between ice growth and melt and ice volume within the Introduction, possibly with a schematic, although this may require removal of a figure elsewhere. We expand on this plan below in response to the reviewer's specific requests.

- I would like to have more information in the method sections on the metrics used: e.g., which are diagnostics (i.e., growth and melt rate terms) vs. derived from the internal temperature profiles (conductive fluxes), and how they relate to the method used to get the in-situ values

from IMBs. This would also help to shift the focus on the use of IMBs to assess CMIP6 models (for a GMD manuscript).

All assessed metrics are direct model diagnostics, including conductive fluxes, and we will make this clear. This differs from the IMB measurements, where conductive fluxes are derived from temperature measurements – but in this case, melt and growth fluxes are themselves also derived, from elevation measurements.

Of course, some of the individual models themselves will derive conductive fluxes from internal temperatures within their sea ice thermodynamics codes – though many derive full implicit solutions, including temperatures and conductive fluxes, in a single thermodynamic solver.

- I find it difficult to fully discuss ice top heat conduction without discussing the snow layer. This is somewhat covered in section 5, but could be mentioned earlier and throughout the analysis.

In CICE5.1.2 and CICE4, the top heat conduction is defined as the downwards heat flux from the snow/ice surface to the interior of the top layer of the snow-ice column. This top layer may be snow, if snow is present, or ice if snow is not present. When top conductive heat fluxes were derived from the IMB data, we strove to match this definition by measuring the conduction to the surface of the snow-ice column (rather than the top surface of the ice, although these are coincident if no snow is present).

- Discussing the mushy layer model results, the authors vaguely refer to un-reported terms in the basal heat balance. This should be more more specific, otherwise I find it somewhat difficult to interpret these model results. I suspect that the authors are referring to the treatment of sea ice congelation not fully accounting for the conductive heat flux in the CICE mushy layer congelation scheme (see Plante et al., 2024), but I am not sure. If it is so, then it is important adapt the discussion accordingly: it is not a "missing diagnostic term", but some flux sent to the ocean during congelation. As it is congelation-related, it could explain the lower growth flux but should not directly affect the melt flux.

Thank you for drawing our attention to this interesting study, which we found particularly valuable in its clear presentation of the CICE mushy-layer scheme. The issue the authors identified with congelation growth not accounting for all basal energy loss may be contributing to the differences seen with these models, and we will reference it accordingly. However, we think that the main cause is something more fundamental.

Eq. (16) of Plante et al. may be particularly relevant:

$$\frac{\partial h_c}{\partial t} = \frac{F_{bot} - F_{cb}}{L\rho_i(1 - \varphi_{init})}$$

which expresses the rate of vertical congelation growth as the basal energy imbalance (numerator) divided by a term proportional to the solid ice fraction $(1 - \varphi_{init})$, which is small by default at 0.15.

What we think this equation will mean practically, is that only quite a small basal energy imbalance will be necessary to produce a given amount of ice *volume* production (due to the small denominator / large liquid fraction). Because ice volume production results in a shallowing of the vertical temperature gradient, and decreased ice conduction, we think this means that basal conduction will be naturally much lower in the mushy-layer scheme than in the BL99 scheme. The 'missing term' referred to is almost certainly the freezing of the liquid

water entrained into the mushy layer. This appears to us to correspond to change in enthalpy of existing ice, and it is not clear to us that this energy budget term is systematically reported by the mushy-layer models, as it does not correspond to a change in elevation diagnosed by the standard CICE5.1.2 volume budget terms.

We will attempt to summarise this concisely.

- I find it a bit confusing that the selected CMIP6 members are referred to as the "IMB sample", given that they are compared to IMB buoys.

Yes, this nomenclature is maybe not the best. We think the reviewer's later suggestion of 'CMIP6 subset' is sensible.

- Figures are often mis-referenced in the text.

Apologies; we will endeavour to correct all instances of this.

Specific points and edits suggestions (some repeating the points above):

- L24: "although some improvement in agreement with reference datasets with model resolution and model complexity is discernible": This wording is not clear, please be more specific.

Most of the quoted studies find that higher resolution models, and models with more advanced sea ice physics, on average compare better to observations. We will make this both clearer and more specific – for example, by explicitly quoting which studies find this.

- L26-32: This paragraph is confusing, mainly because it is so concise that it becomes too vague. It would be worth expanding on these processes so that they are well understood by the reader before getting into the analysis. For instance, thicker ice melts more than thin ice: because of the larger area of ice surviving longer through summer?

We agree that the wording is confusing here, because we are probably unintentionally conflating two issues: seasonal sea ice melt, and sea ice loss in a warming climate. Seasonally, thicker ice melts *less* than thin ice, due to the albedo feedback. However, annual mean sea ice volume loss for a given increase in atmospheric forcing is on average greater if the initial sea ice volume is greater, due to the thickness-growth feedback (which primarily acts in the freezing season). This is the result we are referencing in Holland et al. (2006) and Chen et al. (2023). We will expand on this, along the lines indicated above, and ensure that these two processes are clearly separated.

- L30: it is not clear if "both" refers to the growth and melt, or to the processes.

This means both (ice volume) and (ice growth and melt). We will find a wording that clarifies this.

- L34: This sentence is confusing: it is not clear which climate variables you are referring to, and how it relates to the complexity of sea ice volume processes.

It is in areas such as this that a schematic would probably be most valuable. We will explicitly label those climate variables that are most important for sea ice evolution that we think have not previously been extensively evaluated.

- L39: "Although evaluation of the internal processes of the sea ice is in principle even more difficult": This is a bit subjective, unless there is a reasoning added to this statement.

This is due to the even greater sparsity of observations of ice thermodynamics as compared to surface variables. We will clarify this.

- L42-45: This is also tedious to follow for readers not familiar with this study. I think it is worth adding some precisions.

We will expand on this with examples. One of the key findings of West et al. (2019) was that the ice thickness seasonal cycle of HadGEM2-ES was too amplified; the IMB evaluation showed both ice growth and melt to be much stronger in the model than in the buoy measurements.

- L65: why not simply "CMIP6 subset" instead of IMB? It would be less confusing when discussing models vs IMB observations.

Yes, this is better and we will make this change.

- L68: fast ice growth -> rapid ice growth (to avoid confusion with fast ice)

A good suggestion. Thank you.

- L77: You could cite Bitz and Lipscomb (1999) here.

Also a good suggestion; this will be done.

- Section 2.2.: Are all these IMBs all from CRREL with 10cm vertical resolution, or is there a number of them with higher resolution (e.g., SAMS, SIMB3, etc.)? It would be useful to indicate how they relate/compare to other data (for instance MOSAIC IMBs, etc., e.g. Koo et al., 2020)

Yes, these are all IMBs from CRREL with 10cm vertical resolution, because this was the dataset used in West et al. (2020). As this is primarily a model evaluation paper, we feel it would be out of scope to include new in situ observations, or use these to evaluate the IMB dataset. This is particularly true because converting the IMB data to a form usable for model evaluation was a very substantial task. Use of new buoy data would instead be a valuable study in its own right.

- L116: "was found to demonstrate well" -> displayed? Was used to characterize? It currently sounds like the quality of the observations were assessed against other unnamed references.

It would indeed be accurate simply to say that the IMB dataset displayed seasonal and regional variability, but we are trying to make a stronger statement: that the nature of this variability was consistent with evidence from other data sources. For example, the IMB data suggested a later onset of melting, and earlier cessation of melting, in the North Pole than the Beaufort Sea region; this is consistent with satellite measurements (e.g. Markus et al., 2009). As a second example, there were many more instances of nonzero winter ocean heat fluxes in the North Pole than the Beaufort Sea region; this is consistent with our understanding of the circulation of Atlantic Water within the Arctic Ocean, as the North Pole region is much closer to the AW inflow.

Both these examples are discussed in West et al. (2020), but would probably represent too much detail for the current study. We will expand a little on our point – perhaps say 'The dataset displayed seasonal and spatial variability consistent with observational and theoretical understanding of the Arctic Ocean climate.'

- L121: Please add reference.

Apologies for this oversight, the reference is West et al. (2019) and we will add this.

- L125: Do you mean that you use both the sounders elevation measurements and the temperature profiles to determine the material interface positions? Also, I would like to have a measure of the uncertainties, and how it compares with other methods (e.g., see Richter et al., 2023)

The interface positions were determined using only the elevation measurements. However, there were multiple instances of decreases in surface or snow-ice interface elevation at times when surface melting could not conceivably have been taking place. Hence temperature measurements were used to validate the top melt fluxes: if a decrease in elevation occurred on a day when surface temperature was below a fixed threshold (-2C) the decrease was judged due to some process other than melting (e.g. wind drifting) and the melt flux for that day reset to 0.

In West et al. (2020) we assess uncertainties in IMB-estimated fluxes due to a number of factors, including salinity, conductivity, density and choice of reference layer. We did not explicitly discuss uncertainty due to direct elevation or temperature measurement uncertainty. However, we note that Lei et al. (2014) quoted an accuracy of 0.01m and 0.1K for elevation and temperature measurements from an IMB similar to those used in this study. These values would imply uncertainty more than an order of magnitude smaller than that due to those issues we do explicitly evaluate. We will briefly mention this.

This aside, Richter et al. (2023) was illuminating to read; it's encouraging to see that reasonably accurate ice thickness measurements can be made in the absence of elevation sensors.

- Section 2.3: Some of these data are also based on models, which are somewhat related to the components in some of the CMIP6 members. Could this interfere with the results? For instance, PIOMAS is, to my understanding, based on the POP model, which is also used in some of the CMIP6 members.

This is a good point, although we think that PIOMAS and ERA5 are the only datasets affected by this, and only PIOMAS includes an explicit sea ice model (ERA5 assumes all sea ice to be 1.5m thick).

- L143: "the sea ice state simulation of the IMB subset": rephrase. Perhaps: the sea ice state simulated by the subset models?

Yes, that sounds better. We will rephrase this.

- L144: "we restrict the evaluation"

Yes, this reads better and will be changed.

- L155-156: This could be presented with respect to the PIOMAS and CryoSat-2 seasonal cycle.

Thank you, we will mention as well when these reference datasets display minima and maxima.

- L170 suggestion: "There is strong correlation (0.81) between [...]"

Yes, this is better and we will make this amendment.

- L172-178: This paragraph is a bit difficult to follow. Some revisions would be helpful.

E.g.:

o "We compare the annual mean ice thickness to the anomaly in global 2m air temperature relative to the 1850-1899 average".

Thank you for the suggested rewording, we will make this change.

o Clarify "Arctic Ocean temperature": sounds like ocean temperature. I believe that you rather refer the T2m.

Yes, we do and will clarify this.

- L197: "Figure 2" -> I believe you are referring to Figure 1. Many other figures are also mis-referenced (e.g. Figure 7,8,9)

Yes, this is Figure 1. We will amend the misreferencing.

- Figure 4: Remove the "down" from the labelling of the SW down as it shows upwellingSW. It is also difficult to identify which curves is the net or downwelling radiation as they are not completely staggered (due to the CMCC curves). I recommend moving the net SW in a separate plot.

We will relabel these graphs 'SW radiation' and 'LW radiation', and make it clear in the caption that downwards=positive. We will try to make the net SW / downwelling SW plots more distinct.

- Figure 5: Missing information in the caption. The box plots are distributing the monthly means... for each year in the study period?

Yes, this is averaged across all years in the study period (1985-2014). This will be clarified.

- L246: typo :: or -> for

Thanks for noticing; this will be corrected.

- L263-267 (and also L501-502): I am not sure I understand the 2nd. Are you referring to the fact that IMBs will not sample the new ice forming in leads that form during the observation period?

It's not so much the new ice **in leads** which isn't sampled enough – though it isn't – that would be more comparable with the modelled frazil flux. The problem is thin ice, already formed; this normally grows rapidly, is included in the modelled congelation flux congelation, but is by its nature also insufficiently sampled by the IMBs.

This is primarily an Eulerian-Lagrangian problem. Models report their diagnostics from an Eulerian perspective; they report on the state of all the ice within a specific area, and therefore automatically include the contribution of all rapidly growing thin ice. IMBs report from a Lagrangian perspective as they are advected around with the ice. Hence a rapidly growing new ice floe is sampled only very briefly, by its very nature; soon it thickens to a point at which it is no longer rapidly growing. It contributes only a small amount of rapid growth, or strong conduction, to a monthly mean basal growth or conduction flux.

Another way of thinking about this: even if the IMBs were **deployed** at random, they do not measure a random sample of the sea ice over time, because the probabilities of a particular ice type being sampled by a particular buoy at two different times aren't independent. If at time t the sampled ice is thin and growing rapidly, the probability that the sampled ice is thicker and no longer growing rapidly is greater than the unconditioned probability for times greater than $t+\delta t$, for some relatively small $\delta t$.

- L290: "but display negative conduction in summer". This is interesting. Is this computed from the ice interior, or the conductive flux diagnostic? Is this indicating that the surface temperature is colder than the ice interior despite warmer air temperature?

All conductive fluxes come directly from the models' own reported diagnostics. In this case, a negative top conduction flux would indeed usually indicate that the ice interior is warmer than the ice surface. In the mushy-layer configuration of CICE5.1.2, the top conduction flux forms part of a fully implicit solution for the whole ice and snow column. It is proportional to the difference between the surface skin temperature and the top layer temperature, though the scaling conductivity depends on the (fully prognostic) salinity. We cautiously interpret that, on average, the top layer of the ice-snow column remains warmer than the surface throughout the summer in these models, and will note this in our revision.

It is a counterintuitive result, as atmospheric forcing might be expected to warm the surface to the melting point while the ice interior was still somewhat colder. However, the IMBs are agnostic on this point; some display weakly positive top conductive flux in summer, some weakly negative. It may be related both to the enabling of penetrative solar radiation, and the higher liquid fractions, in these models (such that there is less ice for the radiation to warm to the melting point).

- L310-311 (also L328-330): Please clarify : I do not get how a missing diagnostic term in the energy balance would impact the conduction computed from the simulated temperature profiles. However, as I mentioned above, I believe that you may be referring to the mistreatment of the energy balance in the mushy layer congelation scheme (not only a diagnostic), which results in a wetter and warmer ice base.

It is our understanding that the mushy-layer thermodynamic scheme is often characterised by weaker thermal gradients at the ice base (as energy exchange is inhibited by latent heat transformations), hence smaller basal conductive fluxes are to a degree not surprising – though we were taken aback by the magnitude of the difference.

When we refer to missing terms in the basal energy balance, what we are really referring to is the 'simple' basal energy balance equation (latent heat of fusion * d/dt ice base elevation = ocean heat flux + downwards basal conductive flux). This equation is valid for most thermodynamic ice models, but not the mushy-layer model – because d/dt ice base elevation does not completely characterise the latent heat exchanges that take place at the ice base. 'Missing terms' was possibly not a very good way to describe this. We will try to improve this.

(See also our response to general comment 4, which hopefully makes a similar point with different words).

- L344-345: In the CICE mushy layer scheme, a significant portion of the ice growth happens via frazil formation (DuVivier et al., 2022), again due to the treatment of the conductive flux in the congelation scheme. The fact that the frazil flux is not included here may thus impact the mushy group more than the other models.

We agree that this issue is likely contributing to the difference with these models and will state this, but think it is probably not the whole cause (see our replies to general comment 4, and to L388 comment below).

- Figure 7: In the panel c, the IMB points lie outside of the IMB uncertainty shading... Is that a plotting error?

Yes, the averages for the Beaufort Sea region have been mistakenly plotted here instead of the North Pole (while the shaded areas correctly represent the North Pole). We will correct this.

- L350: Is there more behind this attribution? It is not obvious when looking at Fig. 7 (which is also is referred to as Fig. 8 in the text). One could argue that the relationship is weak even among the model groups (e.g., there is low top melting in the purple models, without much of a slope).

Partly that is an effect of the vertical scale, such that the intermodel variability in top melting is quite hard to see. For example, the mushy-layer models have correlation between top melting an ice thickness of -0.99 (!!) in the North Pole region, which isn't obvious from the figure because the top melting variation is small on this scale. Among the GSI8.1 models correlation is -0.69, still substantially higher than the correlation across the ensemble as a whole (-0.51). The disconnect between whole ensemble and model group correlation is much reduced when looking at total melt rather than top melt, so we think that our point was valid – but probably it should be backed up with numbers given the difficulty of reading this from the graph.

- L358: "overlapping between ice growth and melt terms" -> rephrase: the terms are not overlapping, their season is.

Proposed rewriting: 'any ice growth or melt which occur in the same month are effectively invisible to the ice thickness seasonal cycle'.

- L375 (comment): I think that the fact that the CNRM-CERFACS models also display large conduction indicates that we have here a thermodynamic issue, rather than a diagnostic one (i.e., a problem in diagnostics would not influence the simulated internal temperature profiles).

This seems reasonable, but in the absence of further guidance from the model authors we will not speculate further. It is possible that a diagnostic issue might directly affect multiple output fluxes without affecting the internal thermodynamic evolution. Regardless, the conclusion that the diagnostics themselves are inaccurate is certain, and confirmed by the model authors.

- L388: It is also likely that lower growth is partly associated to the missing frazil contribution, which is more impactful using the current CICE mushy layer scheme.

With respect to the lower basal growth evaluated in Figure 5e,f this is almost certainly the case and we will state this. However note that in Figure 2c ice growth/melt is diagnosed as the amplitude of the ice thickness seasonal cycle; relative to ice thickness, this quantity is also quite low in the mushy-layer models. As calculated, it should include the frazil ice growth term as well as the congelation growth. So frazil/congelation splitting probably isn't the whole story with these models, though it is part of it.

- L389-391: I am not sure I got this right, perhaps it is worth spending more words here to clarify. I.e, the high (outward?) net LW flux is indicative of a cold surface temperature, despite the warmer atmosphere?

The net LW flux, and all radiative and conductive fluxes, are reported using the sign convention downwards=positive. Hence a higher net LW flux in one model than another indicates that the downwelling LW difference is relatively greater in magnitude than that of the upwelling component.

The downwelling component, very roughly speaking, diagnoses the state of the atmosphere, while the upwelling component, as it is determined by the surface temperature, is affected also by the state of the sea ice.

All other things being equal, we would expect an increased downwelling LW flux to alter the surface temperature and increase the upwelling LW accordingly. The relatively greater net LW in the mushy-layer models suggests that there may be a structural factor inhibiting this response to some extent. Reduced conduction through sea ice is one such plausible mechanism.

- L440: "pushing points to the left" -> towards smaller insulance ?

Yes, this is the meaning and we will state this.

- Figure 9: It looks like there is also vertical differences in the distributions. Are the conductive fluxes also computed differently in panel a and b?

No; for each grid point, the computed conductive fluxes are exactly the same. It is the thermal insulance that changes. Hence each insulance class is made up of a different set of points in panel a and b.

For example, a point that has an uncorrected thermal insulance in the 1.5-2 Km2W-1 range might then have a corrected insulance in the 1-1.5 Km2W-1 – so it would transfer from one distribution to another. This means that the bars are showing different distributions of conductive fluxes between panels a and b – because they are made up of different sets of points, even though each individual point has the same conductive flux.

We will try to improve the wording here to make this clearer.

- L466-475: I feel like there is a missing point here: to me, this analysis is indicating that the relationship between ice thickness and top-melt is a large scale one (i.e., it related to the ice Area, via the albedo effect), and thus is not showing when looking at individual grid points and IMBs.

Yes, this is a good point and we will state this. We believe it does not invalidate the main point of this paragraph, which is that the sampling bias of IMBs does not significantly impact the measured top melting fluxes – indeed, the melt-thickness correlation being less visible on the small sale of the IMBs is one reason for this.

- L476: ameliorate -> address?

Better. We will make this change.

References:

Bitz, C. M., and W. H. Lipscomb (1999), An energy-conserving thermodynamic model of sea ice, J. Geophys. Res., 104(C7), 15669–15677, doi:10.1029/1999JC900100.

Koo, Y., Lei, R., Cheng, Y., Cheng, B., Xie, H., Hoppmann, M., Kurtz, N.T., Ackley, S.F., Mestas-Nunez, ˜ A.M., 2021. Estimation of thermodynamic and dynamic contributions to sea ice growth in the Central Arctic using ICESat-2 and MOSAiC SIMBA buoy data. Remote Sens. Environ. 267, 112730. https://doi.org/10.1016/j.rse.2021.112730.

Plante, M., Lemieux, J.-F., Tremblay, L.B., Tivy, A., Angnatok, J., Roy, F., Smith, G., Dupont, F., (2024), Using Icepack to reproduce Ice Mass Balance buoy observations in landfast ice:

improvements from the mushy layer thermodynamics, The Cryosphere, 18, 1685–1708,https://doi.org/10.5194/tc-18-1685-2024.

Richter, M. E., Leonard, G. H., Smith, I. J., Langhorne, P. J., Mahoney, A. R., and Parry, M.: Accuracy and precision when deriving sea-ice thickness from thermistor strings: a comparison of methods, J. Glaciol., 69, 879–898, https://doi.org/10.1017/jog.2022.108, 2023

**References for review response**

Lei, R., Li, N., Heil, P., Cheng, B., Zhang, Z., and Sun, B.: Multiyear sea ice thermal regimes and oceanic heat flux derived from an ice mass balance buoy in the Arctic Ocean, J. Geophys. Res.-Oceans, 119, 537–547, https://doi.org/10.1002/2012JC008731, 2014.

Markus, T., J. C. Stroeve, and J. Miller (2009), Recent changes in Arctic sea ice melt onset, freezeup, and melt season length, J. Geophys. Res., 114, C12024, doi:10.1029/2009JC005436.

West, A., Collins, M., Blockley, E., Ridley, J., and Bodas-Salcedo, A.: Induced surface fluxes: a new framework for attributing Arctic sea ice volume balance biases to specific model errors, The Cryosphere, 13, 2001–2022, https://doi.org/10.5194/tc-13-2001-2019, 2019.

---

## Author Response (AR1)

**Authors' response**

This is an updated response to the reviews of 'CMIP6 models overestimate melt, growth & conduction fluxes relative to ice mass balance buoy estimates', describing changes made in response to the reviews.

**Updated response to Reviewer 1 (John Toole)**

Throughout this response, the original review is reproduced in black, our original responses are shown in red, and updated responses describing changes made are shown in blue.

West and Bockley carry out an evaluation of vertical heat flux and Arctic sea ice evolution in a set of recent climate models with reference to observations from Ice Mass Buoys. The work documents a variety of differences between models and between models and observations and as such, informs model developers what to focus on going forward. I believe with minor revision, the work is appropriate for publication in Geoscientific Model Development.

**We thank the reviewer for their kind remarks.**

The authors make no mention of the possible differences between thermodynamic and mechanical (ice rafting) ice growth. I have been told by ice experts that the thicker ice classes are most certainly created by rafting, not basal growth. Perhaps their focus on the IMB data (that are rarely if ever deployed in really thick floes) makes this point moot. But it might be worth a mention.

While it is certainly possible that any ice floe measured by an IMB could have been created by a rafting or ridging event prior to the initial deployment, we do not believe it is likely that such an event could be misdiagnosed as basal growth during the operation period of an IMB, because such an event would likely cause serious disruption to the the IMB sensors. For example, Perovich et al. (2023) document a ridging event during the MOSAiC campaign which caused a permanent data outage to the under-ice acoustic sensor.

In West et al. (2020), the post-processing of the IMB data is described, as part of which months with sudden step changes in basal elevation are manually identified and removed from the dataset. In fact it was thought that these instances (of which there are four in total) were more likely due to false bottom formation, as in each case there was a step change in the opposite direction several months later, and this process, unlike ridging or rafting, lacks the severe dynamical effects which might prevent further IMB sensor operation.

We will briefly summarise this issue in the paper, to make a concise argument as to why we think IMBs are unlikely to misdiagnose dynamic ice growth as thermodynamic.

On rereading this comment, we think we may have interpreted it too narrowly, and that Reviewer 1 was commenting rather on the fact that our study does not address the ways in which differing ice dynamics can affect modelled ice volume. While we think this is a factor of secondary importance to the atmospheric thermal forcing, it may indeed be of similar importance to the ice thermodynamic choices identified in this study. We have noted this caveat in the Conclusion section – it's beyond the scope of our study, but a reader should be aware that the thermodynamics isn't the whole story. In the similar vein of mechanical influences, I wonder what impacts leads in the ice cover have on model state evolution. In the one modeling paper I did that utilized IMB data (Toole et al., JGR 2010), the summer basal melt was virtually totally accounted for by ocean heat gain by solar radiation into leads. Heat conduction through the ice was of secondary importance.

We thank the reviewer for bringing Toole et al. (2010) to our attention, a very interesting paper which quantifies the difficulty of oceanic heat convergence in significantly contributing to the basal melting of sea ice in summer. This agrees with our own understanding from observations and model results (e.g. Maykut & McPhee, 1995; Steele et al., 2010; Keen and Blockley, 2018). It did not seem plausible that downwards basal conduction could directly contribute to basal melting in any significant way because the ice base is held at the melting temperature and any upwards temperature gradient would be very weak. The basal conduction is of interest mainly in the freezing season, when it is the principal driver of congelation growth.

Hence we think it is likely that the principal driver of intermodel variation in basal melting is ocean-ice heat flux variability, and that this is in turn driven by solar heating variability, which then links back to the ice-albedo feedback. We will note this in our revision.

**We have noted this at the end of the introductory paragraph to the basal conductive flux evaluation.**

This is no doubt beyond the scope of what the authors wish to discuss, but I would have appreciated a few lines explaining how heat flux between ocean and ice is derived in the models. I'm particularly interested in this for the mushy-layer models where I wonder how ice-ocean stress is conducted through a mushy layer.

Although we do not directly evaluate ocean-ice heat flux in this study, we agree that this is of relevance as the ocean-ice heat flux is the principal driver of basal melt in summer. In Section 2 we will briefly summarise how this is formulated, but give a fuller description here. The models are actually quite similar in how they parameterise this flux.

For all of the CICE-based models, the ocean-ice heat flux is computed by a relatively simple formula, based on McPhee (1995):  $F_{bot} = -\rho_w c_w c_h u (T_w - T_f)$ , where  $\rho_w, c_w, c_h, u, T_w$ , and  $T_f$  refer to water density, water heat capacity, heat transfer coefficient, relative ice-ocean speed, water temperature, and freezing temperature respectively. All use a heat transfer coefficient of 0.006. The mushy-layer models differ not in the computation of the ocean-ice heat flux, but in how the heat is transferred upwards into the ice, and indeed in how  $T_f$  is calculated.

In the case of LIM and GELATO, the formulation is similar, although the reference is McPhee (1992) which is itself referenced by the later study; the heat transfer coefficient remains 0.006. The MRI formulation, described in Section 3 of Mellor and Kantha (1989), is conceptually identical, although it gives a theoretical derivation of the heat transfer coefficient in terms of limits and does not explicitly state its eventual value.

**We have noted the common formulation of the models' ocean-to-ice heat flux parameterisation.**

Lastly, going beyond the proposal that future MIPs include more information about heat fluxes, I wonder if the authors might offer thoughts on what improvements to model parameterizations are needed, and perhaps what observations are needed to better identify model shortcomings.

This is a good suggestion and we will implement it. We think that our study directly shows the benefits of modelling penetration of solar radiation into ice. It may also show the benefits of the mushy-layer parameterisation as these models display among the smallest growth/melt biases, but this conclusion would have to be more cautious as the evidence is more indirect and the basal conductive fluxes are possibly a little too low in these models. This is particularly the case given the evidence presented by Reviewer 2 (Mathieu Plante) of overproduction of frazil ice by these models.

We have added a paragraph discussing the implications of our results for future developments in sea ice thermodynamics modelling.

I close with some small issues:

Line 11: might be helpful to detail what specific fluxes are being evaluated. From the following sentence, one might assume the focus is heat fluxes, but ice-ocean, air-ice, air-sea, ???

Thank you, we will attempt to explicitly state evaluated fluxes in this sentence as concisely as possible.

In the end, we stated the evaluated fluxes in the preceding sentence, replacing 'thermodynamic and mass balance diagnostics' with 'fluxes of melt, growth and conduction'.

Line 15: just to check, "realistic" or "unrealistic"? From the context, I am thinking the latter is intended.

No, 'realistic' is intended. For example, models which do not allow solar radiation to penetrate sea ice assume all solar radiation to be either reflected or absorbed. We would expect this to increase top melting and decrease basal melting – and indeed this is what happens, hence 'physically realistic'. However this does not mean that this lack of solar penetration has a real-world counterpart, which is possibly where the confusion arises. Perhaps 'physically consistent' would be a better way of wording this.

This sentence has been restructured to clarify that we mean the differences between evaluated fluxes are physically consistent with differences in thermodynamic parameterisations.

Line 21: similarly, do you intend to say "underestimate" or "overestimate"?

'Underestimate'. We believe this is a fair summary of Figure 1d of Notz et al. (2020), which shows that the rate of change of sea ice area with respect to global mean surface temperature is substantially lower in CMIP6 models than in observations.

**Specific figure reference added.**

Line 26: I don't follow why the mean state and future trend are "closely related." I guess I'm thinking of a positive correlation. If thicker ice melts more than thinner ice, wouldn't a present day overestimate of ice thickness imply a greater decrease (i.e., negative rate of change in ice thickness) over time (a negative correlation)?

A negative correlation between annual mean ice thickness, and its rate of change under climate warming, is indeed what is shown by the referenced studies. It seems to be largely driven by the thickness-growth feedback in the freezing season; as ice is lost, end-of-summer ice area and thickness is lower, leading to stronger growth during the winter.

This contrasts with *seasonal* melt, short-term melting during a single summer, which tends to be higher for thinner than for thicker ice due to the sea ice albedo feedback: a positive correlation.

Reviewer 2 raised a very similar question, along with other concerns over clarity in the Introduction. In response to these, we intend to include a fuller discussion of the links between ice area & volume, and ice growth & melt, and the Arctic climate variables driving these, possibly with a schematic.

We have reworded this paragraph (along with most of the Introduction) to improve the logical flow and describe the processes in greater detail. The large-scale result is described first: that annual mean ice thickness decreases more under increased climate forcing for thicker initial ice. This is then attributed to the thickness-growth feedback, which opposes the effect of the climate warming and which is stronger for thinner than for thicker ice. We then note that the surface albedo feedback partially negates its effect.

Line 53: I don't understand what melt, growth and conduction fluxes are. Are you talking about heat fluxes associated with melt and growth?

Yes. Strictly speaking, melt & growth are fluxes of mass, not energy; in this study, we treat them as interchangeable, related by the latent heat of fusion of water 3.35e5 Jkg-1. We will explicitly state this, and mention circumstances in which this approximation is less accurate (saltwater / briny ice). In the case of the IMBs this is specifically addressed in West et al. (2020), with errors due to salinity uncertainty quantified.

We have clarified this point in the Introduction, and added a couple of sentences describing the relationship between volume, mass and energy fluxes. The difficulties of translating between these quantities were covered in West et al. (2020) for the IMB data; for the models, the difference between these quantities is closely related to the larger discussion of the mushy-layer models and the 'missing terms', which we have referenced.

Top of page 3: are these model quantities available at each model grid point?

The conduction fluxes are produced by the CMIP6 models as means over ice, and are therefore only available over points with sea ice present. The melt and growth fluxes are produced as gridbox means, and are therefore available everywhere, but are zero over points with no sea ice. To ensure comparability with the IMB data and the conduction fluxes, we divide the melt and growth fluxes by ice concentration prior to analysis, to convert to ice-only values.

A paragraph has been inserted at the start of Section 4 to describe the model data processing in more detail; we felt that this was the most appropriate place to give this information.

Line 78: please clarify if the penetrating solar radiation mentioned here is through the ice or into the water within leads.

Through the ice; we will clarify this.

**Amended to explicitly state this.**

Line 110: I think these are "atmospheric" pressure sensors. Aside: absent an in-water pressure sensor at the base of the IMBs, the freeboard of the supporting ice floe cannot be determined by IMBs. All of their reported variations in surface and basal elevations are thus relative to the IMB body.

'Air temperature and pressure' is short for 'air temperature and air pressure', but possibly 'air temperature and sea level pressure' would be a clearer wording here. We are aware that the IMB tracks the vertical Lagrangian motion of the ice floe, as well as the horizontal. In all IMBs with data published by CRREL, the zero level is set to the snow-ice interface at the time of deployment, with all subsequent measurements relative to this level (though the interface itself can change relative to the ice floe through top melting). These issues are discussed in more detail in West et al. (2020).

'Pressure' changed to 'sea level pressure'.

Line 119: what is this "reference layer"? Same as mentioned at the top of page 5?

Yes. The ordering here clearly needs to be changed, and these two statements linked; this will be amended.

The ordering has been changed such that the IMB data processing is now described first – including the definition of the reference layer – and the results and uncertainty analysis summarised after this.

Figure 1: I find the term "Arctic Ocean region" confusing, in part because the label in the figure is over a yellow background region, not blue as noted in the caption. I wonder if calling the North Pole and Beaufort Sea areas "subregions" might help?

We would be happy to make this change, and can move the 'Arctic Ocean region' label so that it is actually on top of the blue shading.

We have changed 'region' to 'subregion' at the point at which these regions are defined, and at the beginning of the model evaluation in Section 4. However, we have not changed every instance of 'region' throughout Section 4 because the word is used very frequently and the meaning is unambiguous by this point.

Line 320: isn't the quantity estimated the time rate of change of the heat content? d/dz (vertical heat flux) equal to d/dt (Heat Content). I don't know what a "heat storage flux" is.

Yes, the heat storage flux is precisely d/dt (heat content). We will clarify this.

We have changed the explanation of ice heat storage to reflect the reviewer's suggested wording.

**References**

Keen, A. and Blockley, E.: Investigating future changes in the volume budget of the Arctic sea ice in a coupled climate model, The Cryosphere, 12, 2855–2868, https://doi.org/10.5194/tc-12-2855-2018, 2018.

Perovich, D. K., Richter-Menge, J. A., Jones, K. F., and Light, B.: Sunlight, water, and ice: Extreme Arctic sea ice melt during the summer of 2007, Geophys. Res. Lett., 35, L11501, https://doi.org/10.1029/2008GL034007, 2008.

Perovich, D., Ian Raphael, Ryleigh Moore, David Clemens-Sewall, Ruibo Lei, Anne Sledd, Chris Polashenski; Sea ice heat and mass balance measurements from four autonomous buoys during the MOSAiC drift campaign. Elementa: Science of the Anthropocene 5 January 2023; 11 (1): 00017. doi: https://doi.org/10.1525/elementa.2023.00017

Steele, M., Zhang, J., and Ermold, W.: Mechanisms of summer Arctic Ocean warming, J. Geophys. Res.-Oceans, 115, C11004, https://doi.org/10.1029/2009JC005849, 2010.

**Updated response to second review (Mathieu Plante)**

Throughout this response, the original review is reproduced in black, our original responses are shown in red, and updated responses describing changes made are shown in blue.

Review of: "CMIP6 models overestimate sea ice melt, growth & conduction relative to ice mass balance buoy estimates" by Alex E. West and Edward W. Blockley.

This manuscript discusses the performance of CMIP6 models in reproducing the sea ice thermodynamics as measured from Ice Mass Balance (IMB) buoys in the central Arctic and the Beaufort Sea. In particular, the authors inter-compare the simulated sea ice growth and melt from different CMIP6 members and discuss differences in terms of the simulated sea ice state (yearly mean thickness and area). They effectively group the selected CMIP6 members according to their component characteristics, allowing them to identify common patterns and the source of discrepancies. They find that in general, the simulated thermodynamic fluxes in the selected CMIP6 members respond in a realistic manner to the simulated climatological sea ice state but overestimate the magnitude of these fluxes.

I find that the manuscript is very interesting, pertinent and of good quality. The presented results are insightful on model sensitivities, and on the importance of an accurate representation of the heat fluxes at the air-ice and ice-ocean interfaces. The use of in-situ observations to evaluate the internal sea ice thermodynamics is also a significant contribution. This analysis relevant for publication in the Journal of Model Developments.

Nonetheless, I find that the more general contributions in the manuscript are sometimes difficult to isolate within the detailed results. There are also a few needed corrections and clarifications to better guide the reader towards the main conclusions.

I thus recommend this manuscript to be accepted for publication, after major revisions, as listed below.

**We thank the reviewer for their positive and constructive remarks.**

**Mathieu Plante**

General comments:

- There is a tendency to use overly concise wordings, at the cost of preciseness and sometimes accuracy. This is especially present in introduction, where some steps in the reasonings are skipped, likely because they seem obvious to the authors, but effectively leaves it to the reader to work it out. I believe it is worth spending more wordings to be more precisions, especially for processes that are later referred to in the analysis.

We understand the point the reviewer is making. We think there is scope for a fuller discussion of the relationship between ice growth and melt and ice volume within the Introduction, possibly with a schematic, although this may require removal of a figure elsewhere. We expand on this plan below in response to the reviewer's specific requests.

We have extensively altered the Introduction, to improve the logical flow and to include more detail about sea ice thermodynamic processes. The arguments described are illustrated very well by an already-published figure (Figure 1 of West et al., 2020, reproduced below, which we have referenced). Because of this, and because the maximum number of figures is already

reached, we have decided not to include a new schematic, and hope that the arguments are now sufficiently clear that this is satisfactory.

**Figure 1 of West et al. (2020)**

- I would like to have more information in the method sections on the metrics used: e.g., which are diagnostics (i.e., growth and melt rate terms) vs. derived from the internal temperature profiles (conductive fluxes), and how they relate to the method used to get the in-situ values from IMBs. This would also help to shift the focus on the use of IMBs to assess CMIP6 models (for a GMD manuscript).

All assessed metrics are direct model diagnostics, including conductive fluxes, and we will make this clear. This differs from the IMB measurements, where conductive fluxes are derived from temperature measurements – but in this case, melt and growth fluxes are themselves also derived, from elevation measurements.

Of course, some of the individual models themselves will derive conductive fluxes from internal temperatures within their sea ice thermodynamics codes – though many derive full implicit solutions, including temperatures and conductive fluxes, in a single thermodynamic solver.

In response to this and a similar question from Reviewer 1, we have added a paragraph at the top of Section 4 explaining in more detail how the evaluated model diagnostics are obtained.

- I find it difficult to fully discuss ice top heat conduction without discussing the snow layer. This is somewhat covered in section 5, but could be mentioned earlier and throughout the analysis.

In CICE5.1.2 and CICE4, the top heat conduction is defined as the downwards heat flux from the snow/ice surface to the interior of the top layer of the snow-ice column. This top layer may be snow, if snow is present, or ice if snow is not present. When top conductive heat fluxes were derived from the IMB data, we strove to match this definition by measuring the conduction to the surface of the snow-ice column (rather than the top surface of the ice, although these are coincident if no snow is present).

**We have amended the introduction to the top conduction paragraph to make its definition clear.**

- Discussing the mushy layer model results, the authors vaguely refer to un-reported terms in the basal heat balance. This should be more more specific, otherwise I find it somewhat difficult to interpret these model results. I suspect that the authors are referring to the treatment of sea ice congelation not fully accounting for the conductive heat flux in the CICE mushy layer congelation scheme (see Plante et al., 2024), but I am not sure. If it is so, then it is important adapt the discussion accordingly: it is not a "missing diagnostic term", but some flux sent to the ocean during congelation. As it is congelation-related, it could explain the lower growth flux but should not directly affect the melt flux.

Thank you for drawing our attention to this interesting study, which we found particularly valuable in its clear presentation of the CICE mushy-layer scheme. The issue the authors identified with congelation growth not accounting for all basal energy loss may be contributing to the differences seen with these models, and we will reference it accordingly. However, we think that the main cause is something more fundamental.

Eq. (16) of Plante et al. may be particularly relevant:

$$\frac{\partial h_c}{\partial t} = \frac{F_{bot} - F_{cb}}{L\rho_i (1 - \varphi_{init})}$$

which expresses the rate of vertical congelation growth as the basal energy imbalance (numerator) divided by a term proportional to the solid ice fraction  $(1 - \varphi_{init})$ , which is small by default at 0.15.

What we think this equation will mean practically, is that only quite a small basal energy imbalance will be necessary to produce a given amount of ice *volume* production (due to the small denominator / large liquid fraction). Because ice volume production results in a shallowing of the vertical temperature gradient, and decreased ice conduction, we think this means that basal conduction will be naturally much lower in the mushy-layer scheme than in the BL99 scheme. The 'missing term' referred to is almost certainly the freezing of the liquid water entrained into the mushy layer. This appears to us to correspond to change in enthalpy of existing ice, and it is not clear to us that this energy budget term is systematically reported to CMIP6 by the mushy-layer models, as it does not correspond to a change in elevation diagnosed by the standard CICE5.1.2 volume budget terms.

We will attempt to summarise this concisely.

We have substantially expanded the discussion of the mushy-layer models' low basal conductive flux in Section 4.2 to discuss both the problem the reviewer raises, and the additional problem that we describe in our reply. We conclude that the basal conductive flux is likely biased low (in magnitude) in these models. This finding is also referenced late in the Conclusion, where we suggest implications for ice thermodynamics modelling after the suggestion of Reviewer 1.

- I find it a bit confusing that the selected CMIP6 members are referred to as the "IMB sample", given that they are compared to IMB buoys.

Yes, this nomenclature is maybe not the best. We think the reviewer's later suggestion of 'CMIP6 subset' is sensible.

All instances of 'IMB subset' have been replaced with 'CMIP6 subset'.

- Figures are often mis-referenced in the text.

Apologies; we will endeavour to correct all instances of this.

We think that we have identified and corrected all instances of misreferencing, particularly the widespread misreferencing of Figure 7 as 8, and 8 as 9.

Specific points and edits suggestions (some repeating the points above):

- L24: "although some improvement in agreement with reference datasets with model resolution and model complexity is discernible": This wording is not clear, please be more specific.

Most of the quoted studies find that higher resolution models, and models with more advanced sea ice physics, on average compare better to observations. We will make this both clearer and more specific – for example, by explicitly quoting which studies find this.

We have expanded this to give explicit descriptions of the findings of Long et al. (2021) and Chen et al. (2023). The described finding of the second paper is perhaps somewhat weaker than that of the first; we have stated this, and this then leads into the next point.

- L26-32: This paragraph is confusing, mainly because it is so concise that it becomes too vague. It would be worth expanding on these processes so that they are well understood by the reader before getting into the analysis. For instance, thicker ice melts more than thin ice: because of the larger area of ice surviving longer through summer?

We agree that the wording is confusing here, because we are probably unintentionally conflating two issues: seasonal sea ice melt, and sea ice loss in a warming climate. Seasonally, thicker ice melts \*less\* than thin ice, due to the albedo feedback. However, annual mean sea ice volume loss for a given increase in atmospheric forcing is on average greater if the initial sea ice volume is greater, due to the thickness-growth feedback (which primarily acts in the freezing season). This is the result we are referencing in Holland et al. (2006) and Chen et al. (2023). We will expand on this, along the lines indicated above, and ensure that these two processes are clearly separated.

As part of our amendments to the Introduction, we have reworded this paragraph to improve the logical flow. First, we state the large-scale, long-term effect: annual mean ice thickness decreases more under increased climate forcing for thicker initial ice. This is then attributed to the thickness-growth feedback, which opposes the effect of the climate warming and which is

stronger for thinner than for thicker ice. We then note that the surface albedo feedback partially negates its effect.

- L30: it is not clear if "both" refers to the growth and melt, or to the processes.

This means both (ice volume) and (ice growth and melt). We will find a wording that clarifies this.

The second half of this sentence has now been removed, and replaced with a longer discussion of the processes driving the relationship between sea ice thickness on the one hand, and sea ice growth and melt on the other hand.

- L34: This sentence is confusing: it is not clear which climate variables you are referring to, and how it relates to the complexity of sea ice volume processes.

It is in areas such as this that a schematic would probably be most valuable. We will explicitly label those climate variables that are most important for sea ice evolution that we think have not previously been extensively evaluated.

We have now listed the climate variables that we view as being particularly important in understanding sea ice state evolution, both in our initial discussion and in this paragraph, when we discuss observational limitations.

- L39: "Although evaluation of the internal processes of the sea ice is in principle even more difficult": This is a bit subjective, unless there is a reasoning added to this statement.

This is due to the even greater sparsity of observations of ice thermodynamics as compared to surface variables. We will clarify this.

This has been stated.

- L42-45: This is also tedious to follow for readers not familiar with this study. I think it is worth adding some precisions.

We will expand on this with examples. One of the key findings of West et al. (2019) was that the ice thickness seasonal cycle of HadGEM2-ES was too amplified; the IMB evaluation showed both ice growth and melt to be much stronger in the model than in the buoy measurements.

This finding has been stated.

- L65: why not simply "CMIP6 subset" instead of IMB? It would be less confusing when discussing models vs IMB observations.

Yes, this is better and we will make this change.

'IMB subset' changed to 'CMIP6 subset'.

- L68: fast ice growth -> rapid ice growth (to avoid confusion with fast ice)

A good suggestion. Thank you.

Change made as suggested.

- L77: You could cite Bitz and Lipscomb (1999) here.

Also a good suggestion; this will be done.

**Citation added.**

- Section 2.2.: Are all these IMBs all from CRREL with 10cm vertical resolution, or is there a number of them with higher resolution (e.g., SAMS, SIMB3, etc.)? It would be useful to indicate how they relate/compare to other data (for instance MOSAIC IMBs, etc., e.g. Koo et al., 2020)

Yes, these are all IMBs from CRREL with 10cm vertical resolution, because this was the dataset used in West et al. (2020). As this is primarily a model evaluation paper, we feel it would be out of scope to include new in situ observations, or use these to evaluate the IMB dataset. This is particularly true because converting the IMB data to a form usable for model evaluation was a very substantial task. Use of new buoy data would instead be a valuable study in its own right.

We have acknowledged the existence of plenty of other IMBs in the introduction to Section 2.2, and given our justification for not using these in the present study.

- L116: "was found to demonstrate well" -> displayed? Was used to characterize? It currently sounds like the quality of the observations were assessed against other unnamed references.

It would indeed be accurate simply to say that the IMB dataset displayed seasonal and regional variability, but we are trying to make a stronger statement: that the nature of this variability was consistent with evidence from other data sources. For example, the IMB data suggested a later onset of melting, and earlier cessation of melting, in the North Pole than the Beaufort Sea region; this is consistent with satellite measurements (e.g. Markus et al., 2009). As a second example, there were many more instances of nonzero winter ocean heat fluxes in the North Pole than the Beaufort Sea region; this is consistent with our understanding of the circulation of Atlantic Water within the Arctic Ocean, as the North Pole region is much closer to the AW inflow.

Both these examples are discussed in West et al. (2020), but would probably represent too much detail for the current study. We will expand a little on our point – perhaps say 'The dataset displayed seasonal and spatial variability consistent with observational and theoretical understanding of the Arctic Ocean climate.'

Proposed wording change made.

- L121: Please add reference.

Apologies for this oversight, the reference is West et al. (2019) and we will add this.

Citation added (already present in references due to a previous citation).

- L125: Do you mean that you use both the sounders elevation measurements and the temperature profiles to determine the material interface positions? Also, I would like to have a measure of the uncertainties, and how it compares with other methods (e.g., see Richter et al., 2023)

The interface positions were determined using only the elevation measurements. However, there were multiple instances of decreases in surface or snow-ice interface elevation at times when surface melting could not conceivably have been taking place. Hence temperature measurements were used to validate the top melt fluxes: if a decrease in elevation occurred on a day when surface temperature was below a fixed threshold (-2C) the decrease was judged due to some process other than melting (e.g. wind drifting) and the melt flux for that day reset to 0.

In West et al. (2020) we assess uncertainties in IMB-estimated fluxes due to a number of factors, including salinity, conductivity, density and choice of reference layer. We did not

explicitly discuss uncertainty due to direct elevation or temperature measurement uncertainty. However, we note that Lei et al. (2014) quoted an accuracy of 0.01m and 0.1K for elevation and temperature measurements from an IMB similar to those used in this study. These values would imply uncertainty more than an order of magnitude smaller than that due to those issues we do explicitly evaluate. We will briefly mention this.

This aside, Richter et al. (2023) was illuminating to read; it's encouraging to see that reasonably accurate ice thickness measurements can be made in the absence of elevation sensors.

We have added a sentence on measurement uncertainty.

- Section 2.3: Some of these data are also based on models, which are somewhat related to the components in some of the CMIP6 members. Could this interfere with the results? For instance, PIOMAS is, to my understanding, based on the POP model, which is also used in some of the CMIP6 members.

This is a good point, although we think that PIOMAS and ERA5 are the only datasets affected by this, and only PIOMAS includes an explicit sea ice model (ERA5 assumes all sea ice to be 1.5m thick).

We have noted that the PIOMAS sea ice model shares many features with the models evaluated in this study, and that this is one reason for supplementing the ice thickness reference with CryoSat-2.

- L143: "the sea ice state simulation of the IMB subset": rephrase. Perhaps: the sea ice state simulated by the subset models?

Yes, that sounds better. We will rephrase this.

Suggested change made.

- L144: "we restrict the evaluation"

Yes, this reads better and will be changed.

Change made.

- L155-156: This could be presented with respect to the PIOMAS and CryoSat-2 seasonal cycle.

Thank you, we will mention as well when these reference datasets display minima and maxima.

This has been added.

- L170 suggestion: "There is strong correlation (0.81) between [...]"

Yes, this is better and we will make this amendment.

Change made.

- L172-178: This paragraph is a bit difficult to follow. Some revisions would be helpful.

E.g.:

o "We compare the annual mean ice thickness to the anomaly in global 2m air temperature relative to the 1850-1899 average".

Thank you for the suggested rewording, we will make this change.

**Change made.**

o Clarify "Arctic Ocean temperature": sounds like ocean temperature. I believe that you rather refer the T2m.

Yes, we do and will clarify this.

Changed to '2m air temperature over the Arctic Ocean'.

- L197: "Figure 2" -> I believe you are referring to Figure 1. Many other figures are also misreferenced (e.g. Figure 7,8,9)

Yes, this is Figure 1. We will amend the misreferencing.

We think that all figure references are now correct.

- Figure 4: Remove the "down" from the labelling of the SW down as it shows upwellingSW. It is also difficult to identify which curves is the net or downwelling radiation as they are not completely staggered (due to the CMCC curves). I recommend moving the net SW in a separate plot.

We will relabel these graphs 'SW radiation' and 'LW radiation', and make it clear in the caption that downwards=positive. We will try to make the net SW / downwelling SW plots more distinct.

Relabelled as suggested; the net SW has been split off into a separate plot, and all line clusters have been labelled in place also.

- Figure 5: Missing information in the caption. The box plots are distributing the monthly means... for each year in the study period?

Yes, this is averaged across all years in the study period (1985-2014). This will be clarified.

The information has been added to the caption.

- L246: typo :: or -> for

Thanks for noticing; this will be corrected.

**Corrected.**

- L263-267 (and also L501-502): I am not sure I understand the 2nd. Are you referring to the fact that IMBs will not sample the new ice forming in leads that form during the observation period?

It's not so much the new ice **in leads** which isn't sampled enough – though it isn't – that would be more comparable with the modelled frazil flux, which we don't attempt to evaluate for this reason. The problem is thin ice, already formed; this normally grows rapidly, is included in the modelled congelation flux congelation, but is by its nature also insufficiently sampled by the IMBs.

This is primarily an Eulerian-Lagrangian problem. Models report their diagnostics from an Eulerian perspective; they report on the state of all the ice within a specific area, and therefore automatically include the contribution of all rapidly growing thin ice. IMBs report from a Lagrangian perspective as they are advected around with the ice. Hence a rapidly growing new ice floe is sampled only very briefly, by its very nature; soon it thickens to a point at which it is no longer rapidly growing. It contributes only a small amount of rapid growth, or strong conduction, to a monthly mean basal growth or conduction flux.

Another way of thinking about this: even if the IMBs were **deployed** at random, they do not measure a random sample of the sea ice over time, because the probabilities of a particular ice type being sampled by a particular buoy at two different times aren't independent. If at time t the sampled ice is thin and growing rapidly, the probability that the sampled ice is thicker and no longer growing rapidly is greater than the unconditioned probability for times greater than t+ $\d